# HPOSS: A hierarchical portfolio optimization stacking strategy to reduce the generalization error of ensembles of models

**Luan Carlos de Sena Monteiro Ozelim**[ORCID]*, **Dimas Betioli Ribeiro**[ORCID], **José Antonio Schiavon, Vinicius Resende Domingues, Paulo Ivo Braga de Queiroz**

Aeronautics Institute of Technology (ITA), São José dos Campos, São Paulo, Brazil

* luanoz@gmail.com

**Data Availability Statement:** All files are available from the Zenodo database (https://zenodo.org/record/8157390).

## Abstract

Surrogate models are frequently used to replace costly engineering simulations. A single surrogate is frequently chosen based on previous experience or by fitting multiple surrogates and selecting one based on mean cross-validation errors. A novel stacking strategy will be presented in this paper. This new strategy results from reinterpreting the model selection process based on the generalization error. For the first time, this problem is proposed to be translated into a well-studied financial problem: portfolio management and optimization. In short, it is demonstrated that the individual residues calculated by leave-one-out procedures are samples from a given random variable $\epsilon_i$, whose second non-central moment is the $i$-th model's generalization error. Thus, a stacking methodology based solely on evaluating the behavior of the linear combination of the random variables $\epsilon_i$ is proposed. At first, several surrogate models are calibrated. The Directed Bubble Hierarchical Tree (DBHT) clustering algorithm is then used to determine which models are worth stacking. The stacking weights can be calculated using any financial approach to the portfolio optimization problem. This alternative understanding of the problem enables practitioners to use established financial methodologies to calculate the models' weights, significantly improving the ensemble of models' out-of-sample performance. A study case is carried out to demonstrate the applicability of the new methodology. Overall, a total of 124 models were trained using a specific dataset: 40 Machine Learning models and 84 Polynomial Chaos Expansion models (which considered 3 types of base random variables, 7 least square algorithms for fitting the up to fourth order expansion's coefficients). Among those, 99 models could be fitted without convergence and other numerical issues. The DBHT algorithm with Pearson correlation distance and generalization error similarity was able to select a subgroup of 23 models from the 99 fitted ones, implying a reduction of about 77% in the total number of models, representing a good filtering scheme which still preserves diversity. Finally, it has been demonstrated that the weights obtained by building a Hierarchical Risk Parity (HPR) portfolio perform better for various input random variables, indicating better out-of-sample performance. In this way, an economic stacking strategy has demonstrated its worth in improving the out-of-sample capabilities of stacked models, which illustrates how the new understanding of model stacking methodologies may be useful.

**Funding:** The authors received grants to cover the APC costs from the Coordination for the Improvement of Higher Education Personnel (CAPES) - Programa de Apoio à Pós-Graduação (PROAP).The funders had no role in study design, data collection and analysis, decision to publish, or preparation of the manuscript.

**Competing interests:** The authors have declared that no competing interests exist.

## Introduction

In general, most scientific applications are related to assessing the relationship between different random entities subjected to a given performance function, $\varphi$. Normally, neither the exact shape of $\varphi$ nor the joint probability density function (pdf) of the random variables are known, requiring scientists to consider numerical approximations to integration problems and statistical estimation techniques for the joint pdf. On the other hand, such methods require the functions involved to be massively evaluated, which comes at a high computational cost.

Some methods used to reduce the number of calls to the performance function have been developed. The most common and relevant ones are those based on surrogate modeling. Surrogate models simulate the input-output relationship established by the performance function. They comprise simplified mathematical models that are much less expensive to evaluate. Among surrogate models, Polynomial Response Surfaces (PRS), Polynomial Chaos Expansion (PCE), Artificial Neural Networks (ANN) and Support Vector Regression (SVR) have been used in the framework of reliability analysis, see [1–8] and references therein. Because of its usual interpolating nature and the straightforward estimation of the prediction's local variance, kriging has also been considered for solving this type of problems. [9–11].

Even though surrogate modeling-based approaches have demonstrated their ability to address complex problems, some tuning issues may impair their efficiency. As a result, selecting the best surrogate model for a given problem remains a difficult task for users [12]. Surrogates are fitted to function values at a set number of points, referred to as the design of experiments (DoE). The surrogates' accuracy is then assessed across the entire domain. Because the fit quality is determined by the data points, choosing models that only minimize a given error metric may result in different results from DoE to DoE, [13].

There are various surrogate models, each based on mathematical assumptions and prior parameter choices. However, no type or tuning is optimal in all circumstances. In some cases, increasing the information in the training set can lead to better models by adaptively increasing the number of sampled points. Active learning methods exist in these cases (for example, by combining Kriging and Monte Carlo Simulation, namely AK-MCS and similar approaches [14, 15]).

On the other hand, in many cases, acquiring new samples of the joint distribution of the input-output random variables is impossible. Some examples are those when the samples come from running costly numerical simulations, or destructive experiments, such as crash simulations which may take 36 h to 160 h to compute a single simulation run [16, 17]. Thus, it is important to enhance the predictive capabilities of the calibrated models without any new training samples.

Aside from acting as surrogates for true models, it is known that combining surrogates' predictions can be an interesting approach to increasing prediction accuracy when compared to individual models [18–21]. In that regard, individual surrogate models combined in the form of a weighted average model can sometimes enhance the accuracy of predictions [13]. This strategy may include surrogates that belong to different analytical classes (different machine learning algorithms, for example), such that this diverse and large set can increase the chances of avoiding poorly fitted surrogates and a DoE dependence on the performance of individual surrogates. This approach is known as the Ensemble of Surrogate Models (ESM).

Literature reveals that ESM approaches can considerably increase the modeling accuracy, as was in the case of wind speed modeling [22]. In the latter work, the authors proposed a novel framework based on the stacking ensemble machine learning method. They considered eleven base machine learning algorithms in several categories (neuron based categories, kernel based, tree based, gradient boosted, least squares boost, curve based, regression based and

hybrid algorithm based) as a first step to then apply a least squares boost using the output of the base algorithms.

In another paper, other authors [23] statistically analyze the generalization error of ensemble learning to assess base-learners' diversity. In their model, they first perform an input feature selection procedure based on various tree-based embedded methods. The candidate models to be stacked (in their case, passed on to a second layer meta-learner) are then selected based on diversity regularization and individual learning capability. Those authors also apply information theory and standard hierarchical clustering algorithms to quantitatively assess the dissimilarity degree among candidate models by analyzing their error distributions. Their stacking ensemble framework employed a two layer-meta learning leave-one-out procedure.

In general, the existing ESM strategies can be split into two groups, namely local ESM and global EM. The latter has unchanged weight factors in the design space, unlike local ESMs. In the present paper, global ESMs will be studied.

In short, a novel stacking strategy shall be presented in the present paper. Such a new strategy comes from reinterpreting the model selection procedure based on the generalization error. For the first time, it is shown that this problem can be translated into a well-studied financial problem: portfolio management and optimization. Such an alternative understanding of the problem allows practitioners to take advantage of established financial methodologies, which can considerably increase the out-of-sample performance of the ensemble of models. A study case is carried out to show the new methodology's applicability. In the next subsections, a few important aspects needed to subsidize the proposition of the new stacking strategy are explored.

## The problem of learning from examples

Let $X$ and $Y$ be two arbitrary sets such that $X$ will be a subset of a $k$-dimensional Euclidean space and $Y$ a subset of the real line. Then, let $\mathbf{x}$ and $y$ be random variables representing the vector of independent variables and the response variable, respectively. Thus, the independent variable will be a $k$-dimensional vector and the response a real number, since $\mathbf{x}$ and $y$ range over the generic elements of $X$ and $Y$. It is assumed that a probability distribution $P(\mathbf{x}, y)$ exists and is defined on $X \times Y$. Despite being unknown, the joint probability distribution $P(\mathbf{x}, y)$ can be written as [24]:

$$P(\mathbf{x}, y) = P(x)P(y|\mathbf{x}) \tag{1}$$

where $P(y|\mathbf{x})$ is the conditional probability (if it exists) of the response $y$ given the independent variable $\mathbf{x}$, and $P(\mathbf{x})$ is the marginal probability of the independent variable [24].

The data set $D_l$, created by sampling $l$ times the set $X \times Y$ according to their joint probability distribution $P(\mathbf{x}, y)$, typically provides examples of this probabilistic relationship. Then:

$$D_l = \{(\mathbf{x}_i, y_i) \in X \times Y\}_{i=1}^{l} \tag{2}$$

When an estimate of the expected value of $y$ is required for an instance of $\mathbf{x}$ that does not appear in the data set $D_l$, a prediction problem is created. Let an estimator be any function $f: X \rightarrow Y$ that is a part of the functional space $\mathcal{F}$. Any estimator will inevitably make some errors because the independent variable $\mathbf{x}$ does not have to be the only factor that influences the response $y$. We will focus on the problem of determining the best estimator given the knowledge of the data set $D_l$, which will be defined as the problem of learning from examples [24].

Suppose one samples $X \times Y$ according to $P(\mathbf{x}, y)$, obtaining the pair $(\mathbf{x}, y)$. Let $\ell(f(\mathbf{x}), y)$ denote the error made when $f(\mathbf{x})$ is predicted instead of $y$ ($\ell$ is the loss function) [12].

Generally, the expected risk of $f$ w.r.t. loss $\ell$ is defined as the expected value of the loss random variable w.r.t. the space $X \times Y$. Mathematically:

$$R_{risk}(f) = \mathbb{E}_{X \times Y}[\ell(f(\mathbf{x}), y)] = \int_{X \times Y} P(\mathbf{x}, y) \ell(f(\mathbf{x}), y) d\mathbf{x} dy \tag{3}$$

where $\mathbb{E}_Z[\cdot]$ denotes the expectation w.r.t. $Z$. If the loss is chosen as the squared difference, it can be shown that the optimal solution which minimizes $R_{risk}(f)$ is given when [25]:

$$f(\mathbf{x}) = f_0(\mathbf{x}) = \int_Y y P(y|\mathbf{x}) dy \tag{4}$$

Other loss functions would result in different optimal solutions, but such choices do not impact the main rationale of the present paper. This comes from the fact that, for whatever loss function is chosen, it would be always possible to assess the quality of the optimal solution by studying the risk defined in Eq (3).

In general, it can be stated that, regardless of the loss chosen, an optimal function will exist. This function, hereby denoted as $f_0(\mathbf{x})$, belongs to $\mathcal{F}$ and will be approximated by another function, $g(\mathbf{x})$, which belongs to a generic subset of $\mathcal{F}$ whose elements are parametrized by some parameters proportional to a given integer $n$, hereby called $\mathcal{G}_n$. Moreover, it is assumed that the sets $\mathcal{G}_n$ form a nested family, that is $\mathcal{G}_1 \subset \mathcal{G}_2 \subset \mathcal{G}_3 \subset \ldots \mathcal{G}_n$. For example, $\mathcal{G}_n$ could be the set of polynomials in one variable of degree $n - 1$ [24].

We could determine which component of $\mathcal{G}_n$ is best for accurately modeling $f_0$ by taking potential functions and using the expected risk as a criterion. Any prior knowledge of the unknown probability distribution $P(\mathbf{x}, y)$ should be considered when defining $\mathcal{G}_n$.

Considering the example set $D_l$, the problem of learning from examples can now be reformulated as the problem of reconstructing the regression function $f_0$ using such a set. In general, the target function $f_0$ can be said to belong to a general class of functions called $\mathcal{F}$. Noisy data is obtained as $(\mathbf{x}, y)$ where $\mathbf{x}$ has the distribution $P(\mathbf{x})$ and for each $\mathbf{x}$, $y$ is a random variable with mean $f_0(\mathbf{x})$ and distribution $P(y|\mathbf{x})$. If one assumes that the noise is additive, one could write:

$$y = f_0(\mathbf{x}) + \eta \tag{5}$$

where $\eta$ is zero-mean with distribution $P(y|\mathbf{x})$.

If the expected risk in Eq (3) were known, the learning problem would be straightforward to solve, as the regression function could be computed by finding the risk's minimum in $\mathcal{G}_n$. This is not true in general, since $P(\mathbf{x}, y)$ and $R_{risk}(f)$ are unknown. The data set $D_l$, which consists of $l$ independent random samples of $X \times Y$ drawn using $P(\mathbf{x}, y)$, is the only source of information. The empirical risk $R_{emp}(f)$ can be used to approximate the expected risk in Eq (3) using this data set $D_l$:

$$R_{emp}(f) = \frac{1}{l} \sum_{i=1}^{l} \ell(f(\mathbf{x}_i), y_i). \tag{6}$$

One is concerned with reducing the expected risk $R_{risk}(g)$ over the set $\mathcal{G}_n$. Given that the candidate function has a finite number of parameters, the optimal strategy would be to minimize the loss function over the set $\mathcal{G}_n$, which would produce the estimator $g_n$ as:

$$g_n = \arg \inf_{g \in \mathcal{G}_n} R_{risk}(g) \tag{7}$$

However, because the data are finite and the functional space $\mathcal{G}_n$ is limited (by, for example, taking into account a set of parametrizations of continuous functions), the only option is to reduce the empirical risk $R_{emp}(g)$ and obtain the function $\hat{g}_{n,l}$ as the final estimate [24]. By using the squared difference as the metric to measure the distance between $\hat{g}_{n,l}$ and the ideal solution $f_0$, the generalization error $G_{error}$ can be defined as follows:

$$G_{error} = \mathbb{E}_X[(f_0 - \hat{g}_{n,l})^2] = \int_X P(\mathbf{x})(f_0 - \hat{g}_{n,l})^2 d\mathbf{x} \tag{8}$$

The generalization error is primarily caused by two factors: The regression function $f_0 \in \mathcal{F}$, which has an infinite number of dimensions, is being approximated by the parametrized function $g_n \in \mathcal{G}_n$, which has a finite number of parameters. The quantity $E[(f_0 - g_n)^2]$, which is the squared distance between the best function in $\mathcal{G}_n$ and the ideal regression function, is used to measure this error, which is known as the approximation error. It is important to note that the approximation error depends only on the class $\mathcal{G}_n$'s approximating power and not on the data set $D_l$.

Another source of error stems from the fact that one minimizes the empirical risk $R_{emp}$ rather than the expected risk $R_{risk}$ to obtain $\hat{g}_{n,l}$. Thus, the estimation error appears and is calculated as $|R_{emp} - R_{risk}|$. Further details can be seen in [24].

It is possible to think of the generalization error as having both a random component, represented by the estimation error and a deterministic component, represented by the approximation error:

$$f_0(\mathbf{x}) = \hat{g}_{n,l}(\mathbf{x}) + \epsilon, \tag{9}$$

such that $E[\epsilon^2] = G_{error}$ and $E[\epsilon] = \mu$.

Combining Eqs (5) and (9):

$$y = \hat{g}_{n,l}(\mathbf{x}) + \epsilon + \eta \tag{10}$$

**Estimating the generalization error by leave-one-out cross validation.** Leave-one-out cross-validation is a variant of cross-validation in which the number of folds is equal to the number of instances in the dataset [26]. The candidate model's prediction error is calculated for each value in the observed dataset using all other values as a training set and the chosen value as a single-item test set. The leave-one-out error $R_{loo}$, which is meant to be an "almost" unbiased (in the sense of [27]) estimate of the generalization error $G_{error}$ [28], was introduced in various contexts in the late 1960s, including those discussed in [27, 29–31]. For the case of the empirical risk presented in Eq (6), the leave-one-out risk estimator (LOOE) can be defined as:

$$R_{loo}(f) = \frac{1}{l}\sum_{i=1}^{l}\ell(f^i(\mathbf{x}_i), y_i). \tag{11}$$

where $f^i(\mathbf{x}_i)$ denotes the model $f$ calibrated on the training set obtained by removing the point $(\mathbf{x}_i, y_i)$ from $D_l$, an then evaluated at $\mathbf{x}_i$.

The asymptotic capabilities of LOOE as a proxy for the generalization error have been studied previously [32], and it has been shown both theoretically and empirically that the leave-one-out error, whenever the learning algorithms are stable in the sense of [28, 33], is a proper proxy. The present paper considers the LOOE a true proxy for generalization error estimates.

For a squared loss, the generalization error for a given model $i$ can be expressed as:

$$G_{error,i} = \frac{1}{l}\sum_{i=1}^{l}(y_k - \hat{g}_{n,l,i,k}(\mathbf{x}_k))^2 \qquad (12)$$

where $y_k$ is the true response at a given point $\mathbf{x}_k$ and $\hat{g}_{n,l,i,k}(\mathbf{x}_k)$ is the value predicted at $\mathbf{x}_k$ by using the model approximated by the parameterized function $g_n$ and calibrated from all the DoE points in $\mathcal{D}_l$ except the data pair $(\mathbf{x}_k, y_k)$.

The previous literature covers the basic concepts of the learning process from a statistical point of view. In the next subsections, the idea of combining several candidate functions (surrogate models) to build the parameterized function $g_n$ shall be discussed.

## Surrogate models

Surrogate models can improve effectiveness and lower the computational costs of a problem or design process. Various surrogate-modeling techniques have been applied to uncertainty analysis, sensitivity analysis, and optimization to create a statistical model of the simulation model. This allows the repeated simulation "runs" to be completed using statistical surrogates in seconds. [34].

As will be presented in the Materials and Methods section, this paper considers various machine learning techniques, assessing their potential use as surrogates. Kriging is a popular form of Gaussian process regression, which we also included in our set of possible surrogate algorithms. Besides the Machine Learning techniques, Polynomial Chaos Expansions will also be considered.

**Polynomial Chaos Expansions—PCE.** Let an input-output model be represented by a function $y = M(x)$, where $x \in \mathrm{R}^n$, $y \in \mathrm{R}^m$, and $n$ is the number of input quantities and $m$ the number of outputs. For simplicity, the $m = 1$ case will be considered in the following description. Both $x$ and $y$ can be described as random variables $X = (X_1, X_2, X_3, \ldots, X_n)$ and $Y$, respectively, due to the uncertainties in the input variables and their propagation to the output [35–37]. For a specific value of $x$, a deterministic algorithm normally computes the corresponding response $y$. The joint pdf of the random vector $X$ is denoted by $f_X$. Assuming that the input random variables $X_i$ are independent, then $f_X$ is a multiplication of the marginal probabilities, $f_X(x) = \prod_{i=1}^{n} f_{X_i}(x_i)$. A polynomial Chaos Expansion (PCE) approximates the response $Y$ as a linear combination of orthonormal polynomials [2].

In a full PCE, the number of expansion factors $NP$ depends on the polynomial order $p$ and the number of random input parameters $n$, being given by $NP = \frac{(n+p)!}{n!p!}$. Also, in the context of multivariate basis of polynomials, they can be constructed as tensor products of univariate orthonormal polynomials which are closely related to the pdfs $f_{X_i}(x_i)$ [36]. For example, in the case of uniform distributions, Legendre polynomials are the ideal basis function. For Normal random variables, on the other hand, Hermite polynomials are of interest [36].

After defining the univariate polynomial basis, regression analysis can compute the PCE coefficients in a non-intrusive and cost-effective manner. The regularized least squares optimization involved in the regression procedure can be solve by different methods, each of which will assume extra penalizations or constraints to the optimization problem. Naturally, different choices will provide different polynomial coefficients and, therefore, different surrogate models.

## Stacking strategies

In work by Wolpert [38], the general idea of machine learning stacking was discussed: for a given set of predictors, instead of selecting a single one from this set (in a winner-takes-all fashion), a more accurate predictor can be obtained by combining all (or most of) the predictors in the set.

Breiman [39], on the other hand, discussed the concepts behind stacking regressions, which is a method for forming linear combinations of different predictors to give improved prediction accuracy. Thus, suppose we have $m$ different candidate models $\hat{g}_{n,l,i}$, then, in general, the stacked model $\hat{g}_{stk}(\mathbf{x})$ can be obtained as:

$$\hat{g}_{stk}(\mathbf{x}) = \sum_{i=1}^{m} w_i \hat{g}_{n,l,i}(\mathbf{x}) \tag{13}$$

where $w_i$ are real values representing weights. The same rationale discussed in [38, 39] had been previously proposed by Stone [31] and called a "modelmix".

Breiman [39] also discusses that a stacking regression strategy as described in Eq (13) has two main issues. The first is that since each candidate model was constructed using the training data, obtaining $w_i$ by minimizing the squared error over this same training data will be prone to overfitting, which implies that generalization will be poor.

The leave-one-out cross-validation data can be used to diminish this issue, as noted in both [38, 39]. On the other hand, the second issue is more challenging. Since all candidate models attempt to predict the same phenomenon, they are typically highly correlated. The $w_i$ produced will be extremely sensitive to even the smallest changes in the data if a straightforward least-squares reduction of the errors is performed. Generalization will again be inadequate. According to Breiman [39], using ridge regression would be preferable to estimate the regression coefficients of strongly correlated variables. More discussion on these two issues can be found in [40].

Since the proposed methodology is based on an economic portfolio optimization approach, general remarks on this topic are presented in the next section.

## Portfolio optimization: A financial stacking strategy

The most frequent financial issue is probably portfolio creation. Investment managers must create portfolios considering their opinions and projections of risks and returns. Markowitz studied this topic and indicated that different levels of risk correspond to distinct optimal portfolios in terms of risk-adjusted returns [41].

Allocating all the investments to assets with the highest predicted returns is rarely the best course of action. Instead, to create a diversified portfolio, one should consider the correlations across various investments [42]. In this regard, several works have explored portfolio optimization procedures, especially using machine learning techniques. A complete review can be found in [43].

### Modern Portfolio Theory—MPT

The main statistical basis for Markowitz's proposition is that whenever assets' returns have negative Covariance Cov, the Variance, Var of their linear combination is less than the weighted sum of their Variances. Mathematically:

$$\mathrm{Var}(aX + bY) = a^2\,\mathrm{Var}(X) + b^2\,\mathrm{Var}(Y) + 2ab\,\mathrm{Cov}(X, Y) \tag{14}$$

This indicates that portfolios with less risk can be obtained for a fixed target portfolio return by properly selecting the assets to combine. Despite Markowitz's theory's simplicity and apparent robustness, some practical issues show up when considering portfolios that are only built by gathering assets based on minimizing portfolio variance (given specific returns). The optimization routine will generate very different portfolios if the expected returns deviate slightly from the forecasted future values [44]. While disregarding the forecasting process for the returns improves things, it does not resolve the instability problems. The rationale is that positive-definite covariance matrices must be inverted to use quadratic programming methods (all eigenvalues must be positive). When the covariance matrix is numerically ill-conditioned, i.e., has a high condition number, this inversion is vulnerable to significant mistakes [45]. As a result, different solutions have been studied to the portfolio construction problem, a few of which are described in the next subsections.

### Hierarchical portfolio construction

The Hierarchical Risk Parity (HPR) was proposed in [42] to address three major concerns of quadratic optimizers, in general, and Markowitz's critical line algorithm (CLA), in particular: instability, concentration, and underperformance. Based on the data in the covariance matrix, HPR uses contemporary mathematics (graph theory and machine-learning techniques) to create a diversified portfolio. Contrary to quadratic optimizers, HPR does not need the covariance matrix to be invertible and Monte Carlo studies demonstrate that HPR produces lower out-of-sample variance than CLA. Compared to conventional risk parity methodologies, HPR generates less risky portfolios out-of-sample.

The HPR justification is based on the observation that the covariance matrix of the portfolio's asset returns may be visualized as a full graph. Conversely, this method suggests that simpler and more relevant hierarchies are concealed within such comprehensive graphs, which oversimplifies hierarchies. Then, HPR applies a hierarchical clustering technique to the covariance matrix as part of an unsupervised learning strategy. The HPR methodology then indicates recursively re-allocating risk over the assets after identifying asset clusters. An inverse-variance portfolio is constructed when the hierarchy is determined, and the cluster variances are computed [42].

The inverse-variance portfolio is the one whose weights minimize the portfolio variance whenever the covariance matrix is diagonal. Thus, in this case, each asset is weighted in inverse proportion to its returns variance [42]. Pure inverse-variance approaches have already been explored as a stacking strategy for machine learning in [46].

### Robust optimization

Some interesting approaches try to account for the fact that the returns observed are samples of the random variables involved and only present a glimpse of their real behavior. Therefore, the values obtained cannot be taken as deterministic and must be treated in an uncertain framework. The Python package *RSome* [47, 48] provides a full framework for implementing these approaches.

The present paper considers the portfolio construction problem with a robust optimization approach introduced in [49]. The robust model is presented as:

$$
\begin{aligned}
\max \min_{z \in \mathcal{Z}} \quad & \sum_{i=1}^{n}(p_i + \delta_i z_i)x_i \\
\text{s.t.} \quad & \sum_{i=1}^{n} x_i = 1 \\
\text{s.t.} \quad & x_i \geq 0, \forall i
\end{aligned}
\tag{15}
$$

where the affine term $p_i + \delta_i z_i$ represents the random stock return, and the random variable $Z$ (whose samples are $z_i$) is between $[-1, 1]$, so the stock return has an arbitrary distribution in the interval $[p_i - \delta_i, p_i + \delta_i]$. For simplicity, it is assumed in the present paper that $\delta_i = \sqrt{\text{Var}(\epsilon_i)}/2$. The uncertainty set $\mathcal{Z}$ is given as:

$$\mathcal{Z} = \{z : ||z||_\infty \leq 1, ||z||_1 \leq \Gamma\} \tag{16}$$

and $\Gamma$ is the budget of uncertainty parameter.

Often, financial assets can be clustered prior to their combination in a portfolio. This can enforce diversity of assets, which is of utmost interest to investors. Thus, the next subsection explores this concept.

## Clustering returns of financial assets

The clustering of financial assets' returns can be done using hypothesis testing frameworks, or may encompass hierarchical concepts.

**Nonparametric hypothesis tests.** A hypothesis test for equality of distribution can be used as a first step in determining how different random variables are when each sample is compared. Generally speaking, these tests will create a statistical framework to examine the degree to which two or more samples differ from two or more random variables.

Let $X_1$ and $X_2$ be the continuous random variables underlying two populations of interest, and $F_1$ and $F_2$ be their respective distribution functions. The general system of hypotheses when one compares two populations is:

$$H_0 : X_1 \overset{d}{=} X_2 \quad \text{against} \quad H_1 : X_1 \overset{d}{\neq} X_2 \tag{17}$$

where $X_1 \overset{d}{=} X_2$ means that $F_1(t) = F_2(t) \forall t \in \mathbb{R}$ and $X_1 \overset{d}{\neq} X_2$ means that $\exists A \subset \mathbb{R} : F_1(t) \neq F_2(t), t \in A$ with $Pr(A) > 0$.

Because they do not establish any assumptions about the distribution of each random variable being compared, nonparametric tests are especially appealing when the types of random variables studied are unknown. Two of theses tests are explored in detail.

- Kolmogorov-Smirnov Test

The Kolmogorov–Smirnov test may test whether two underlying one-dimensional probability distributions, with densities $F_1(x)$ and $F_2(x)$, respectively, differ. In this case, the Kolmogorov–Smirnov statistic $KS$ is the supremum of the absolute difference between densities [50]. One drawback of this test is that it may be ineffective for the equality of distribution assessment. Some claim that the Cucconi test can be more effective for that [51].

- Cucconi test

For the Cucconi test:

$$F_i(t) = G\left(\frac{t - \mu_i}{\sigma_i}\right), \quad i = 1, 2 \tag{18}$$

where $G(\cdot)$ is the distribution function for a continuous variable with location 0 and scale 1, $\mu_i$ is the location of population $i$, and $\sigma_i$ is its scale. Let observations $X_{i1}, \ldots, X_{1n_i}$ be random samples from population $i$. For the location-scale problem, Cucconi [52] proposed a rank test based on:

$$C = \frac{U^2 + V^2 - 2\rho UV}{2(1 - \rho^2)} \tag{19}$$

where

$$U = \frac{6 \sum_{i=1}^{n_1} W_{1i}^2 - n_1(n+1)(2n+1)}{\sqrt{n_1 n_2 (n+1)(2n+1)(8n+11)/5}} \tag{20}$$

and

$$V = \frac{6 \sum_{i=1}^{n_1} (n+1-W_{1i})^2 - n_1(n+1)(2n+1)}{\sqrt{n_1 n_2 (n+1)(2n+1)(8n+11)/5}} \tag{21}$$

$n = n_1 + n_2$, $W_{ji}$ denotes the rank of $X_{ji}$ in the pooled sample $X = (X_{11}, \ldots, X_{1n_1}, X_{21}, \ldots, X_{2n_2}) = (X_1, \ldots, X_{n_1}, X_{n_1+1}, \ldots, X_n)$ and $\rho = 2(n^2 - 4)/((2n+1)(8n+11)) - 1$. Under $H_0$, $\mathbb{E}[U] = \mathbb{E}[V] = 0$ and $Var(U) = VAR(V) = 1$ [51]. The p-values associated with such tests can be calculated by bootstrap techniques [51].

**Directed Bubble Hierarchical Tree clustering.** In a completely unsupervised and deterministic manner, hierarchical clustering algorithms enable discovery of relationships and structures within datasets. The Directed Bubble Hierarchical Tree (DBHT) uses the topological property of the PMFG (Planar Maximally Filtered Graph) to find the clustering [53].

The PMFG is a generalization of the Minimum Spanning Tree (MST) included in the PMFG as a subgraph. It is created using the same steps as the MST, except that the weaker planarity condition is used instead of the non-loop condition (i.e., each added link must not cut a pre-existent link). The PMFG can retain more links and information than the MST because of this less severe topological constraint. It can be demonstrated, in particular, that each PMFG contains precisely $3(N-2)$ links.

The DBHT uses the topological structure of the PMFG to identify a clustering partition for each node in it [53]. The traditional agglomerative clustering process is then used to obtain an entire hierarchical structure (dendrogram) between and within clusters.

Normally linkage algorithms analyze the sorted list of distances $D_{i,j}$ between nodes $i$ and $j$ and construct the dendrogram by compiling subsets of candidate models with the smallest distances; the clustering is then obtained from the dendrogram after pre-selecting the "number of clusters" desired. Instead, the DBHT reverses this process: first, the clusters are identified using topological analysis of the planar graph, and then the hierarchy is built between and within the clusters. Therefore, the distinction between the traditional agglomerative clustering process and DBHT involves the type of information used and the methodology.

In a recent study [54], researchers quantified the amount of information on stock return correlations filtered by various hierarchical clustering methods. Their findings demonstrate that the DBHT can perform better than other methods by retrieving more data with fewer clusters. Additionally, they demonstrate how, depending on the clustering method, the economic information is hidden at various levels of the hierarchical structures.

The DBHT algorithm considers a dissimilarity (distance) and a similarity matrix. In essence, both matrices are required because the PMFG is a weighted graph, and weights are typically similarity measures for edges (a larger weight of an edge corresponds to a stronger similarity between the connected nodes). The edges are also associated with a distance or, more generally, a non-negative dissimilarity measure [53]. Therefore, it is important to present some candidates for these matrices.

- Pearson correlation distance

Considering the works of [53, 54], an interesting distance metric can be defined in terms of Pearson's correlation coefficients $\rho_{ij}$ between pairs of assets. This way, a $m \times m$ distance matrix, whose elements are $D_{ij} = \sqrt{(1 - \rho_{ij})/2}$, can be defined.

- Kendall's $\tau$ correlation distance

Similarly to the Pearson correlation distance, it is possible to consider Kendall's $\tau$ rank correlation to build a distance metric $\tau_{ij}$ between pairs of assets. Literature reveals that this correlation metric better captures co-movements, while compared to Pearson correlation, especially in the realm of clustering financial time series [55].This way, a $m \times m$ distance matrix, whose elements are $D_{\tau,ij} = \sqrt{(1 - \tau_{ij})/2}$, can be defined.

- Spearman's $\rho$ correlation distance

Literature also indicates that Spearman's $\rho$, $\rho_S$, has a good performance when used to cluster financial assets' returns [55] and, therefore, can be used to build a distance metric $\rho_{s,ij}$ between pairs of assets. This way, a $m \times m$ distance matrix, whose elements are $D_{\rho,ij} = \sqrt{(1 - \rho_{s,ij})/2}$, can be defined.

- Generalization error distance

It is possible to define the distance between two algorithms $i$ and $j$ by computing the absolute difference between the generalization error of each one. Thus, let $\Gamma(i, j)$ denote this difference, then:

$$\Gamma(i,j) = |G_{error,i} - G_{error,j}| \tag{22}$$

- Relative Kullback–Leibler divergence as a distance

Consider the problem of comparing two approximate distributions, $V$ and $S$, using a third reference pdf, $P$. Using the KL divergence to calculate the absolute value of the difference between the KL divergences of both $V$ and $S$ concerning the same function $P$ is possible. Thus [56]:

$$\mathcal{D}_P(V||S) = |D(P||V) - D(P||S)| \tag{23}$$

where $D(P||V)$ is the KL divergence between $P$ and $V$, defined as [57]:

$$D(P||V) = \int_{-\infty}^{\infty} p(x)\log\left(\frac{p(x)}{v(x)}\right) dx \tag{24}$$

in which $p(x)$ and $v(x)$ are the densities of $P$ and $V$, respectively.

Thus, from Eqs (23) and (24):

$$\mathcal{D}_P(V||S) = \left| \int_{-\infty}^{\infty} p(x)\log\left(\frac{p(x)}{v(x)}\right) dx - \int_{-\infty}^{\infty} p(x)\log\left(\frac{p(x)}{s(x)}\right) dx \right| \tag{25}$$

$$= \left| \int_{-\infty}^{\infty} p(x)\log\left(\frac{s(x)}{v(x)}\right) dx \right| \tag{26}$$

An interesting choice for the reference pdf is the Dirac delta pseudo-distribution. Such pseudo-distribution can be modeled as the limit of a Normal random variable with mean and

variance tending to zero. Thus, in the case where the reference random variable is a Dirac delta centered at $\mu_{ref}$, it can be seen that the generalized metric in Eq (25) becomes:

$$D_{\delta,\mu_{ref}}(V||S) = \left| \int_{-\infty}^{\infty} \delta(x - \mu_{ref}) \log\left(\frac{s(x)}{v(x)}\right) dx \right| = \left| \log\left(\frac{s(\mu_{ref})}{v(\mu_{ref})}\right) \right| \tag{27}$$

Finally, for a zero-centered Dirac delta pseudo-distribution, it can be seen that the distance metric adopted could be:

$$D_{\delta,0}(V||S) = \left| \log\left(\frac{s(0)}{v(0)}\right) \right| \tag{28}$$

Eq (28) indicates that calculating the distance metric $D_{\delta,0}(V||S)$ reduces to the calculation of the density ratio between both distributions at the origin. The estimated density ratio function can be used in many applications, such as the inlier-based outlier detection [58] and covariate shift adaptation [59]. Other useful applications for density ratio estimation were summarized in [60].

Some python implementations of the RuLSIF (Relative unconstrained Least-Squares Importance Fitting) method can estimate the alpha-relative density ratio by minimizing the squared loss between the true and estimated alpha-relative ratios. This method is detailed in [58, 61].

Literature indicates that the density ratio problem can be approached by multidimensional densities via $k$-nearest-neighbor distances [62], by a probabilistic classification [63] or even an infinitesimal classification [64]. A general discussion can be found in [63].

• Generalization error similarity

For this case, it is possible to express the similarity of two datasets based on their Generalization error distance defined in Eq (22) as $S_{G_{error},i,j} = \exp\left(-\frac{\Gamma(i,j)}{\max_{i,j}\Gamma(i,j)}\right)$. Now that a brief review of the literature has been presented, the Material and Methods considered in the present paper shall be presented

## Material and methods

The main contributions of the present paper can be split into two: a theoretical and an applied one, hereby named the Hierarchical Portfolio Optimization Stacking Strategy (HPOSS). The theoretical contribution relies on reinterpreting the stacking of surrogate models as the construction of a portfolio of financial assets, indicating how this new understating can result in novel stacking strategies. The applied contribution, on the other hand, is related to the proposition of a new two-step methodological approach to the stacking problem based on well established financial techniques. Such applied contribution shall be illustrated by a study case. Since the theoretical contribution does not rely on any other concepts than the ones described in the Introduction, the following subsections will focus on the tools needed to the develop the HPOSS.

### Methodological steps—HPOSS

The methodological steps presented in Fig 1 should be followed to apply the Hierarchical Portfolio Optimization Stacking Strategy (HPOSS). In general, HPOSS considers two major steps: filtering models worth stacking and calculating the weights for those models. It is, therefore, a

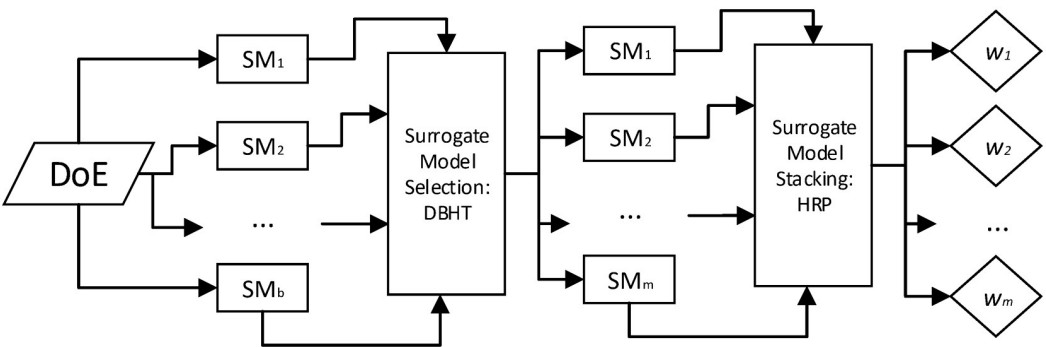

**Fig 1. Methodological steps.**

stacking strategy that necessarily is preceded by a filtering step. Besides, it is understood that many candidate models are considered to account for different analytical families.

From Fig 1, one needs to gather samples of the joint probability distribution of the inputs and outputs. This can be achieved by a DoE. Then, a total of $b$ surrogate models are calibrated by using the DoE. From the calibration process, the samples of the leave-one-out residues are obtained for each surrogate model, which are nothing but samples from the random variables $\epsilon_i$, for $i = 1, \ldots, b$.

**Surrogate modeling.** Given that the main benefit of using surrogate models is to significantly reduce the time needed to perform in-depth analyses of a particular problem of interest, techniques that can produce predictions quickly after the surrogate has been trained are of special interest [34]. In the present paper, to show the applicability of the HPOSS, two classes of surrogates were considered: Machine Learning and Polynomial Chaos Expansion ones.

• Machine Learning Algorithms

We have picked a few of the most popular machine-learning techniques for our study. Using the Python *scikit-learn* library, similarly to [65], the authors gathered some of the available regressor-type estimators by importing the method "all_estimators" from "sklearn.utils" and then discarding some regressors which were not considered of interest. Such a pre-selection process was based on the authors' previous experience calibrating the sklearn models, especially considering the time each algorithm takes to perform the learning process. This way, in order to enforce diversity, a total of forty algorithms gathered into ten model classes were considered, namely [66]: **ensemble** (which try to enhance the overall generalizability and robustness while compared to a single predictor by merging the forecasts of multiple predictors generated using a particular learning method—'AdaBoostRegressor', 'BaggingRegressor', 'ExtraTreesRegressor', 'GradientBoostingRegressor', 'HistGradientBoostingRegressor', 'RandomForestRegressor'); **linear_model** (operate under the assumption that the desired output will be a result of combining the input features in a linear manner—'BayesianRidge', 'ElasticNet', 'ElasticNetCV', 'GammaRegressor', 'HuberRegressor', 'Lars', 'LarsCV', 'Lasso', 'LassoCV', 'LassoLars', 'LassoLarsCV', 'LassoLarsIC', 'LinearRegression', 'LinearSVR', 'OrthogonalMatchingPursuit', 'OrthogonalMatchingPursuitCV', 'PassiveAggressiveRegressor', 'PoissonRegressor', 'QuantileRegressor','RANSACRegressor', 'Ridge', 'RidgeCV', 'SGDRegressor', 'TweedieRegressor'); **tree** (acquire simple decision rules from the provided input data to construct a model capable of predicting the target variable's value—'DecisionTreeRegressor', 'ExtraTreeRegressor'); **dummy** (make predictions using uncomplicated rules, serving as basic benchmarks for comparison with other regression models

—'DummyRegressor'); **gaussian_process** (also known as kriging algorithms—'GaussianProcessRegressor'); **neighbors** (assign a numerical value to a given point by calculating the mean (or other summarizing function) of the values from its closest neighbors—'KNeighborsRegressor'); **kernel_ridge** (combines ridge with the kernel trick—'KernelRidge'); **neural_network** (simple multi-layer perceptron regressor—'MLPRegressor'); **svm** (support vector machine based models—'NuSVR', 'SVR') and **compose** (meta-estimator to regress on a transformed target—'TransformedTargetRegressor'); For further details on each method, we refer the reader to the key references cited in [66], and also the work of [34].

When dealing with a vast list of machine learning models, it becomes evident that hyperparameter tuning may not be imperative [67]. This is due to the fact that when weak predictors are aggregated and combined, they possess the capability to produce robust estimates. By harnessing the collective knowledge of multiple models, even those with relatively modest individual predictive capabilities, the resultant ensemble is able to offset individual shortcomings and attain more dependable and precise predictions. As a result, the emphasis transitions from fine-tuning hyperparameters for individual models to the creation and combination of diverse models, thereby enhancing the ensemble's overall performance and capacity to generalize. This is precisely the focus of the present paper.

- Polynomial Chaos Expansion

Many methods are available to generate the univariate polynomial basis for particular pdfs $f_{X_i}(x_i)$. One is the so-called three terms recurrence method, which is generated using Stieltjes [68] and Golub-Welsch method [69]. This is a considerably stable method, but do not work on dependent distributions. Other approaches include the one discussed in [36], known as the Askey scheme. In the present paper, PCE were fitted to the data using the three terms recurrence method implemented in the *chaospy* [70] algorithm.

- Making predictions with surrogate models

When a candidate surrogate model is calibrated, one wants to perform inferences for new input values. Point estimates are not the best alternative in this case since there are some errors due to sampling and the modeling procedure itself, which must be accounted for during the inferencing process.

If a given surrogate model is chosen, its analytical formulation is known and equal to $r(.)$. Let's consider that the $\hat{\beta}$ parameters have been obtained from a given set of training samples. Then, it is possible to predict the expected value of the response variable given a new set of inputs $x_0$, as $r(x_0)$.

Normally, it is assumed that the relation between the calibrated surrogate model and the output response is subjected to an additive zero-mean noise, which implies that the final estimation of the expected value of the output is exclusively dependent on the function $r(.)$ and on the estimated parameters $\hat{\beta}$. This is a point estimate which may be misleading if solely analyzed. Interval estimations bring much more information, especially in risk analysis assessments, where the most extreme situation are sought.

It is possible, then, to consider the confidence interval of the expected value $r(x_0)$. In this case, one is interested in predicting the mean response, the average response value for a given input $x_0$. To calculate such a confidence interval, one needs to account for the variance in $\hat{\beta}$, which may arise from the training sample sampling process.

On the other hand, if one is interested in predicting the specific output for a particular input $x_0$, then, to estimate the variance of such prediction, it is necessary to consider not only the uncertainty about $\hat{\beta}$, as in the case of the confidence of expectation interval but also the

uncertainty related to our actual prediction. In this case, one is not interested in the mean response (expected value) but in a specific future value.

Overall, the major difference between the confidence of expectation and prediction intervals is that the former accounts for uncertainty in the sampling of the training samples, while the latter accounts for both this sampling issue as well as for the uncertainty of the model prediction itself (i.e., considers the variability of the possible outputs around the predicted mean). In general, prediction intervals are more suited for applications where one is not simply worried about the mean response. Calculating the prediction interval can be achieved by a bootstrap procedure.

A non-parametric estimation technique called Bootstrap was developed by Efron [71, 72] and enables one to calculate the confidence interval (in the statistical sense) for a given statistic of interest. The Bootstrap method is a statistical inference technique that relies only on currently available data (sample). One of its main features is the method's lack of dependence on any consideration of the relevant random variables.

The Bootstrap technique can estimate the sampling distribution of a particular statistic (for example, the sample's mean and variance) by considering that the sample is representative of the population from which the latter has been collected and that the observations are independent and identically distributed.

Thus, to calculate the $1 - \iota$ confidence level prediction interval at a given value of the input value $x_0$, one can use the following bootstrap routine (simplified from the one presented in [73]):

- From the training dataset $\{W_1, W_2, \ldots, W_k\}$, draw a new sample $\{W_1^*, W_2^*, \ldots, W_k^*\}$ of size $k$ with replacement. Each $W_i$ (or $W_i^*$, by consequence) is a pair with an input variable $x_i$ (or $x_i^*$) and an output value $y_i$ (or $y_i^*$).

- Train the surrogate model $r(.)$ with the dataset $\{W_1^*, W_2^*, \ldots, W_k^*\}$ to obtain $r_j(.)$.

- Draw one element at random, $e^{j,*}$, from the errors sample $\{e_1^{j,*}, e_2^{j,*}, \ldots, e_k^{j,*}\}$ such that $e_i^{j,*} = y_i^* - r_j(x_i^*)$.

- Calculate the estimated output value as the sum $r_j(x_0) + e^{j,*}$.

- Repeat $B$ times the last four steps to obtain a bootstrap sample of outputs $\{r_1(x_0) + e^{1,*}, r_2(x_0) + e^{2,*}, \ldots, r_B(x_0) + e^{B,*}\}$.

- From the sample $\{r_1(x_0) + e^{1,*}, r_2(x_0) + e^{2,*}, \ldots, r_B(x_0) + e^{B,*}\}$, calculate the confidence interval for the mean value for a $1 - \iota$ confidence level.

After the surrogates have been trained, it is interesting to discuss further how to cluster the candidate surrogate models, which is the topic of the next sub-subsection.

**Clustering.**   Clustering candidate models corresponds to clustering their performances during the prediction tasks. As indicated in the present paper, the performance of each model will be assessed based on the random variables $\epsilon_i$, whose samples are obtained as the leave-one-out residues. Thus, studying how to group datasets according to a given clustering hypothesis is important.

Applying a clustering algorithm to gather similar surrogate models makes it possible to select $m$ models worth stacking. The selected models are the ones that belong to the cluster that contains the method with minimum observed generalization error (i.e., minimize Eq (12)), which would be the chosen model in a winner-takes-all approach.

In the present paper, the DBHT is used to explore how candidate models can be clustered according to their performance (generalization errors) and analytical similarity (Pearson, Kendall's $\tau$ and Spearman's $\rho$ correlations of leave-one-out residues). This allows one to filter, from a possibly large set of candidate models, the ones who could bring better quality to the stacking

strategy. This choice of similarity and distance metrics is based on the fact that, for the DBHT algorithm, the similarity defines the bubbles (general structure of the graph), which are then hierarchically organized by the distance metric (inter and intra structure of the clusters). By using the generalization error similarity and choosing the cluster containing the minimum observed generalization error, one chooses the models with lower observed errors. On the other hand, using the Pearson correlation distance, the internal hierarchy of the bubbles is such that similar models (same analytical capabilities) are clustered together (because their residues are highly correlated).

The python package *Riskfolio-Lib* [74] presents a numerical implementation of the DBHT method, which shall be slightly modified (adapted to produce different plots) and used in the present paper.

The procedure proposed in [75], where the ratio of densities is calculated based on the ratios of their empirical cumulative distributions, is considered in the present paper to calculate the distance in Eq (28). In short, the empirical cumulative distribution is built by combining simple Heaviside step functions and piece-wise linear approximations. Then, such function is numerically differentiated using a finite difference approach, therefore obtaining an estimate of each density function involved [75].

An alternative expression, which considers that the random variables involved are approximately Normal, can be obtained as [56]:

$$D_{\delta,0}(V||S) \approx \left| \frac{\mu_v^2}{\sigma_v^2} - \frac{\mu_s^2}{\sigma_s^2} - \log\left(\frac{\sigma_s}{\sigma_v}\right) \right| \tag{29}$$

where $\mu_s$, $\sigma_s$ and $\mu_v$, $\sigma_v$ are the means and standard deviations of the random variables $S$ and $V$, respectively. Then, the second crucial step is considered: a stacking strategy is applied to obtain the weights $w_i$, $i = 1, \ldots, m$ of the selected models.

**Stacking.**   In the HPOSS context, the stacking strategy to be considered is the HPR financial approach. This approach is considered with the same distance metric used in the DBHT algorithm, i.e., Pearson correlation distance. Such choice ensures that similar models cannot dominate the weight distribution (avoid concentration), since the weights are calculated based on the global and local hierarchy instead of single model performance. For example, if two models are withing the same cluster, they will share a certain portion of total weight (ascribed to the whole cluster) instead of dominating the weight distribution. This privileges analytical diversity, as desired.

To assess the suitability of the HPOSS, in the study case, just two previously published stacking strategies are explored, both because they have shown to be superior to other simpler alternatives and because they explore the concept of generalization error minimization, which is discussed under different premises in the present paper. The heuristic formulation proposed by Goel et al. in [18] and the optimized weight factor of Acar and Rais-Rohani in [19] are the known stacking strategies selected.

• Heuristic proposed by Goel et al.

The strategy of selecting weights proposed by Goel et al. [18] is formulated as follows:

$$\begin{aligned}
E_i &= \sqrt{G_{error,i}} = \sqrt{\frac{1}{N}\sum_{k=1}^{N}(y_k - \hat{g}_{n,l,i,k})^2} \\
E_{avg} &= \frac{\sum_i E_i}{m}, \omega < 1; \beta < 0 \\
w_i &= \frac{w_i^*}{\sum_i w_i^*}, w_i^* = (E_i + \omega E_{avg})^\beta
\end{aligned} \tag{30}$$

where $m$ is the number of candidate models, $y_k$ is the true response at a given point $\mathbf{x}_k$, $\hat{g}_{n,l,i,k}$ is the prediction from the $i^{th}$ surrogate model calibrated from all the DoE except the data pair $(\mathbf{x}_k, y_k)$ and $N$ is the size of the DoE. The parameters $\omega$ and $\beta$ should be specified beforehand. For instance, $\omega = 0.05$ and $\beta = -1$ is used in the work of [18]. A study of the effect of those parameters has also been performed [18].

- Optimization problem proposed by Acar & Rais-Rohani

The weight factors are here solutions for the minimization of a global error. The influence of the error metric choice is studied in [76]. In their original paper, Acar & Rais-Rohani [19] selected the generalized mean square error concerning the ensemble of models, $GMSE_{\hat{g}_{stk}}$. This metric is defined by:

$$GMSE_{\hat{g}_{stk}} = \frac{1}{N} \sum_{k=1}^{N} (y_k - \hat{g}_{stk,k})^2 \tag{31}$$

where $y_k$ is the true response at a given point $\mathbf{x}_k$ and $\hat{g}_{stk,k}$ is the value predicted by using a stacked model calibrated from all the DoE points except the data pair $(\mathbf{x}_k, y_k)$. The weight factors are the solutions to the following optimization problem:

$$\begin{aligned} \min_{w} \quad & GMSE_{\hat{g}_{stk}} \\ \text{s.t.} \quad & \sum_{i=1}^{m} w_i = 1 \end{aligned} \tag{32}$$

By changing the optimization problem in Eq (32) into a matrix form, Viana et al. [13] obtained an explicit solution. On the other hand, their solution is based on building a matrix based on the average cross-errors of each model obtained by leave-one-out cross-validation.

The problem in Eq (32) can be solved as it is, in an unbounded manner, and also can be changed to a bounded alternative, where the weights are considered to lie within the (0, 1) interval. Both approaches (bounded and unbounded) will be considered in the present paper.

The stacking strategies presented by Goel et al. and by Acar & Rais-Rohani will be used as benchmarks to test the capabilities of the proposed new stacking strategy. Other novel heuristic approaches will be also proposed in the present paper.

In summary, for the HPOSS, low generalization error is obtained by using the DBHT algorithm with a generalization error similarity. High model diversity is achieved by combining the DBHT algorithm and the HPR stacking strategy, both with the Pearson correlation distance metric. One could argue that it would be possible to directly calculate the weights using the HPR rationale on the global/local hierarchy obtained by the DBHT algorithm. This would be possible, but in the present paper, we applied such algorithms sequentially (instead of in a nested manner). To assess the suitability of the HPOSS, a few new general stacking alternatives that encompass statistical ideas will be proposed.

## Results and discussions

As mentioned earlier, the present paper encompasses two main contributions: a theoretical and an applied one. Thus, the results and discussions will be presented according to these two categories.

## Theoretical contribution

As indicated, for a loss represented by the squared difference, $R_{loo}$ can be interpreted as the generalization error and, therefore, as the second central moment of the random variable $\epsilon$. This way, it is possible to consider that the individual residues calculated by leave-one-out procedures are samples from $\epsilon$, which are squared and expressed as their mean by $G_{error}$ following Eq (8).

From this point on, it is considered that the leave-one-out residues, given as $y_i - f^i(\mathbf{x}_i)$ are samples from the random variable $\epsilon_i$.

The main purpose of the present paper is to propose a stacking methodology based solely on assessing the behavior of the random variables $\epsilon$. In this case, for each model to be stacked, there will be a specific random variable $\epsilon_i$ which expresses the residues from its leave-one-out procedure. A robust and accurate stacking strategy is proposed by linearly combining such variables.

By considering a stacking of candidate models as in Eq (13), by assuming that the weights of the models are $w_i$ such that $\sum_{i=1}^{m} w_i = 1$ (as suggested by some authors to have an unbiased response prediction [12]), the following theorem holds:

**Theorem 1**. *The generalization error ($G_{stacked}$) of the stacked model $\hat{g}_{stk}(\mathbf{x})$ is bounded by the moments of the random variable $\sum_{i=1}^{m} w_i \in_i$ as:*

$$\mathrm{Var}\left[\sum_{i=1}^{m} w_i \epsilon_i\right] + \left(\mathbb{E}_X\left[\sum_{i=1}^{m} w_i \epsilon_i\right]\right)^2 = G_{stacked} \tag{33}$$

*Proof.* From Eq (9):

$$
\begin{aligned}
\mathbb{E}_X\left[\left(\sum_{i=1}^{m} w_i \epsilon_i\right)^2\right] &= \mathbb{E}_X\left[\left(\sum_{i=1}^{m} w_i (f_0(\mathbf{x}) - \hat{g}_{n,l,i}(\mathbf{x}))\right)^2\right] \\
&= \mathbb{E}_X\left[\left(\sum_{i=1}^{m} w_i f_0(\mathbf{x}) - \sum_{i=1}^{m} w_i \hat{g}_{n,l,i}(\mathbf{x})\right)^2\right] \\
&= \mathbb{E}_X\left[\left(\sum_{i=1}^{m} w_i f_0(\mathbf{x})\right)^2\right] + \mathbb{E}_X\left[\left(\sum_{i=1}^{m} w_i \hat{g}_{n,l,i}(\mathbf{x})\right)^2\right] - \\
&\quad 2\mathbb{E}_X\left[\left(\sum_{i=1}^{m} w_i f_0(\mathbf{x})\right)\left(\sum_{i=1}^{m} w_i \hat{g}_{n,l,i}(\mathbf{x})\right)\right]
\end{aligned}
\tag{34}
$$

In the case, one considers the joint estimation of the functions $\hat{g}_{n,l,i}(\mathbf{x})$, it can be stated that $w_i f_0(\mathbf{x})$ are the same realizations of the random variable $f_0(\mathbf{x})$ times a constant $w_i$. This leads to

the fact that $\sum_{i=1}^{m} w_i f_0(\mathbf{x}) = f_0(\mathbf{x})$ since $\sum_{i=1}^{m} w_i = 1$. Thus,

$$
\begin{aligned}
\mathbb{E}_X\left[\left(\sum_{i=1}^{m} w_i \epsilon_i\right)^2\right] &= \mathbb{E}_X[(f_0(\mathbf{x}))^2] + \mathbb{E}_X\left[\left(\sum_{i=1}^{m} w_i \hat{g}_{n,l,i}(\mathbf{x})\right)^2\right] - \\
&\quad 2\mathbb{E}_X\left[(f_0(\mathbf{x}))\left(\sum_{i=1}^{m} w_i \hat{g}_{n,l,i}(\mathbf{x})\right)\right] \\
&= \mathbb{E}_X\left[\left(f_0(\mathbf{x}) - \sum_{i=1}^{m} w_i \hat{g}_{n,l,i}(\mathbf{x})\right)^2\right] \\
&= G_{stacked}
\end{aligned}
\tag{35}
$$

where the last line of Eq (35) follows from the combination of Eqs (8) and (13).

Another way to see that, in fact, $w_i f_0(\mathbf{x})$ are the same realizations of the random variable $f_0(\mathbf{x})$ times a constant is to notice that $f_0(\mathbf{x}) = \mathbb{E}_Y[Y|X]$. Thus, as soon as the input variables are sampled, a specific realization of $X$ is known and, therefore, $\mathbb{E}_Y[Y|X]$ is no longer a random variable, but a simple realization.

By considering Eq (35) and using the definition of the Variance of a random variable $Z = h(X)$ as $\mathrm{Var}(Z) = \mathbb{E}_X[Z^2] - (\mathbb{E}_X[Z])^2$, Eq (33) is obtained.

It can be seen from Eq (33) that studying how the random variable $\sum_{i=1}^{m} w_i \epsilon_i$ behaves brings significant information on the capabilities of the stacking strategy. It is possible to notice that this random variable will be a proxy for the actual behavior of the stacked model since minimizing the generalization error implies minimizing both the variance and the expected value of $\sum_{i=1}^{m} w_i \epsilon_i$. Another important conclusion can be drawn from the following Theorem:

**Theorem 2**. *Minimizing the expected value of $\sum_{i=1}^{m} w_i \epsilon_i$ will minimize the expected value of the difference between the best possible regressor $f_0(\mathbf{x})$ and the prediction of the stack of models since*:

$$
\mathbb{E}_X\left[\sum_{i=1}^{m} w_i \epsilon_i\right] = \mathbb{E}_X\left[f_0(\mathbf{x}) - \sum_{i=1}^{m} w_i \hat{g}_{l,n,i}(\mathbf{x})\right]
\tag{36}
$$

*Proof.* From Eq (9):

$$
\mathbb{E}_X\left[\sum_{i=1}^{m} w_i \epsilon_i\right] = \mathbb{E}_X\left[\sum_{i=1}^{m} w_i (f_0(\mathbf{x}) - \hat{g}_{n,l,i}(\mathbf{x}))\right]
\tag{37}
$$

Thus, by noticing that in a joint estimation of the functions $\hat{g}_{n,l,i}(\mathbf{x})$ the values of $w_i f_0(\mathbf{x})$ are the same realizations of the random variable $f_0(\mathbf{x})$ times a constant, Eq (36) follows.

Both Theorems 1 and 2 offer a very attractive alternative to define the weights of each candidate model for the stacking procedure. Also, the literature indicates that the best results are obtained when the stacking strategy can combine the confidence (and not just the predictions) of the lower-level models [77]. Studying not only point estimates, but the random variables $\sum_{i=1}^{m} w_i \epsilon_i$ does precisely that: combines the predictions, represented by $\mathbb{E}_X[\sum_{i=1}^{m} w_i \epsilon_i]$, to the confidence of such predictions, represented by $Var[\sum_{i=1}^{m} w_i \epsilon_i]$.

Another way of interpreting Eq (33) is to acknowledge that it precisely presents the bias-variance trade-off dilemma one encounters while calibrating any model [78]. The bias is represented by $\mathbb{E}_X[\sum_{i=1}^{m} w_i \epsilon_i]$, and the variance, by $Var[\sum_{i=1}^{m} w_i \epsilon_i]$.

**Direct optimization stacking approach.** In our interpretation, each of the $m$ candidate models represents a risky asset whose returns are given as $\epsilon_i$, for $i = 1, \ldots, m$. Therefore, we

aim to find the weights $w_i$, for $i = 1, \ldots, m$, such that the generalization error of the stacked model $\sum_{i=1}^{m} w_i \epsilon_i$ is minimized. In the present sub-subsection novel analytical and numerical solutions are presented for some optimization stacking approaches. Besides, novel heuristic approaches are described. Overall, using all theses alternatives is of interest to show how the adequacy and robustness of the methodology hereby proposed.

- Unbounded weights

Theorem 1 indicates that this problem is reduced to minimizing the sum of the variance and the squared expected value of the random variable $\sum_{i=1}^{m} w_i \epsilon_i$. Thus, let $\mathbf{w}$ be a $m \times 1$ column vector whose components are the weights of each model. Also, let $\mu$ be a $m \times 1$ column vector whose components are the expected values of each candidate model. By denoting $\Sigma$ as the covariance matrix of the random variables $\epsilon_i$, $i = 1, \ldots, m$, then, the minimization problem can be represented as:

$$\min_{\mathbf{w}} \quad \mathbf{w}^T \Sigma \mathbf{w} + \left( \mathbf{w}^T \mu \right)^2$$
$$\text{s.t.} \quad \mathbf{w}^T \mathbf{e} = 1 \tag{38}$$

where $\mathbf{e}$ is a $m \times 1$ column vector whose entries are all equal to one.

Let us consider that the weights $\mathbf{w}$ minimize the problem in Eq (38) are $\mathbf{w}_P$, such that $\mu_P = \mathbf{w}_P^T \mu$. Thus, this same vector of weights will also minimize the following problem:

$$\min_{\mathbf{w}} \quad \mathbf{w}^T \Sigma \mathbf{w}$$
$$\text{s.t.} \quad \mathbf{w}^T \mu = \mu_P$$
$$\text{s.t.} \quad \mathbf{w}^T \mathbf{e} = 1 \tag{39}$$

On the other hand, MPT brings an interesting insight into the optimization problem presented in Eq (39). This results in the determination of the values of weights of portfolios that belong to the so-called Efficient Frontier of assets selection.

The optimization problem in Eq (39) can be analytically solved by using Lagrange multipliers [79], leading to the variance of the portfolio $\sigma_P^2$ being expressed in terms of $\mu_P$ as:

$$\sigma_P^2 = \frac{1}{\lambda_4} \left( \lambda_1 \mu_P^2 - 2\lambda_2 \mu_P + \lambda_3 \right) \tag{40}$$

where it is assumed that $\Sigma$ is non-singular such that its inverse, $\Sigma^{-1}$, exists and:

$$\lambda_1 = \mathbf{e}^T \Sigma^{-1} \mathbf{e}$$
$$\lambda_2 = \mathbf{e}^T \Sigma^{-1} \mu$$
$$\lambda_3 = \mu^T \Sigma^{-1} \mu$$
$$\lambda_4 = \lambda_3 \lambda_1 - \lambda_2^2 \tag{41}$$

Now, the next step is to combine Eqs (33) and (40) to solve the optimization problem in Eq (38). Thus:

$$\frac{1}{\lambda_4} \left( \lambda_1 \mu_P^2 - 2\lambda_2 \mu_P + \lambda_3 \right) + \mu_P^2 = G_{stacked} \tag{42}$$

To obtain real-valued solutions to the quadratic Eq (42), it suffices to observe that:

$$\frac{4\lambda_2^2}{\lambda_4^2} - 4\left(\frac{\lambda_1 + \lambda_4}{\lambda_4}\right)(\lambda_3 - G_{stacked}) \geq 0 \tag{43}$$

which leads to:

$$G_{stacked} \geq \lambda_3 - \frac{\lambda_2^2}{\lambda_4(\lambda_1 + \lambda_4)} \tag{44}$$

Thus, the minimum generalization error of the stacked model is obtained when Eq (44) is equality. Also, for this specific value, the generalization error circle only touches the Efficient Frontier once, precisely at the point where:

$$\begin{aligned} \mu_P &= \frac{\lambda_2}{\lambda_1 + \lambda_4} \\ \sigma_P^2 &= \lambda_3 - \frac{\lambda_2^2(\lambda_1 + 2\lambda_4)}{\lambda_4(\lambda_1 + \lambda_4)^2} \end{aligned} \tag{45}$$

From the values obtained in Eq (45), it is possible to explicitly obtain the $m \times 1$ vector of optimum weights $\mathbf{w}_{opt}$ by defining a $m \times 2$ matrix $K = [\mu, \mathbf{e}]$, a $2 \times 1$ vector $\omega = [\mu_P, 1]^T$, a $2 \times 2$ matrix $A = K^T \Sigma^{-1} K$ and:

$$\mathbf{w}_{opt} = \Sigma^{-1} K A^{-1} \omega \tag{46}$$

It is worth noticing that the optimization problem in Eq (38) is a portfolio management reinterpretation of the optimization problem presented in Eq (32) and explored in [19]. Therefore, the solution in Eq (46) is nothing but an explicit exact solution to Eq (32), given differently and more directly than the one presented in [13].

• Graphical interpretation

The graphical interpretation of the present optimization solution is presented in Fig 2. Such interpretation directly results from the mathematical rationale behind the formulas presented.

First and foremost, Eqs (38) and (39) indicate that minimizing the generalization error of the stacked model implies that the selected portfolio belongs to the efficient frontier. Secondly, Theorem 1 states that the generalization error is nothing but the squared radius of a circle whose origin is at (0, 0) and touches the point $(\sigma_P, \mu_P)$, where $\mu_P$ and $\sigma_P$ are the bias and squared root variance of the stacked model generalization error random variable.

This way, finding the portfolio with the least generalization error translates into finding the intersection of the smallest-radius circle centered at (0, 0) which touches the efficient frontier line. It is interesting to notice from Fig 2 that the winner-takes-all approach, where the model with smallest generalization error is chosen, can provide considerably higher generalization errors when compared to a stacked model.

**Optimization with positive-defined weights.** An important observation of Breiman [39] is that, in general, imposing that the weights are non-negative, i.e., $w_i \geq 0, \forall i$, provides predictors which, almost always, have lower prediction error than the single predictor having lowest cross-validation error.

Also, from a Bayesian perspective, Bayesian models can be weighted by their marginal likelihood. This is known as Bayesian Model Averaging [80–82]. This rationale could also be used for surrogate models, as described in [12]. This implies that the weights represent probabilities, so they should be non-negative. The marginal likelihood is extremely sensitive to the

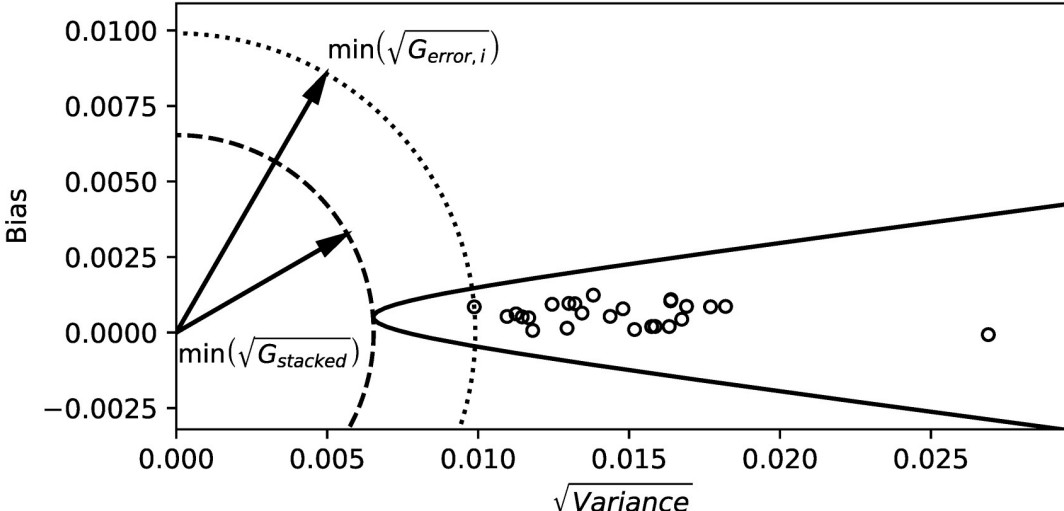

**Fig 2. Markowitz frontier, minimum generalization error asset, and portfolio.**

specification of the prior, whereas parameter estimation is not, and computing the marginal likelihood is typically a difficult process, so while this is theoretically appealing, it is troublesome in practice [83]. Therefore, this Bayesian technique shall not be explored in the present paper.

The new optimization problem would be:

$$\min_{\mathbf{w}} \quad \mathbf{w}^T \Sigma \mathbf{w} + \left(\mathbf{w}^T \mu\right)^2$$

$$\text{s.t.} \quad \mathbf{w}^T \mathbf{e} = 1 \tag{47}$$

$$\text{s.t.} \quad w_i \geq 0, \forall i \tag{48}$$

Such a problem cannot be directly solved using Lagrange's multipliers. Markowitz created the Critical Line Algorithm (CLA), a quadratic optimization technique for situations involving inequality-constrained portfolio optimization. This algorithm stands out because it cleverly gets around the Karush-Kuhn-Tucker requirements and ensures that the precise answer is obtained after a known amount of iterations [42]. This algorithm's description and open-source implementation can be found in the literature [45].

The optimization procedure to solve Eq (47) consists of two steps: first, the minimum variance portfolio with bounded weights is obtained using the CLA algorithm. Then, such weights are used as initial values for the optimization procedure in Eq (47), which can be solved using the Sequential Least Squares Programming (SLSQP) algorithm of *scipy* [84]. This two-step approach is important to make the optimization algorithm look for weights around the tip of the inequality-constrained Markowitz frontier. If the bias of the minimum variance portfolio

is small, there is a good chance that this will be the portfolio which also minimizes the problem in Eq (47).

A special case to Eq (47) happens when the optimization procedure considers the constraint that the mean value of $\sum_{i=1}^{m} w_i \epsilon_i$ is zero. This problem is hereby called *ZeroMinStdWeights* and will be explored in the applications section.

**Novel heuristic approaches.**   As alternative optimization approaches, some heuristic procedures are introduced in the present paper. The rationale behind these approaches is to enforce some sort of regularization during the optimization process, which would provide more robust solutions.

Overall, the basic principle hereby considered is that residues of errors of the portfolio of models, obtained as $\sum_{i=1}^{m} w_i \epsilon_i$, will be approximately distributed as a Normal random variable with zero mean and standard deviation given by $\sqrt{\mathrm{Var}(\sum_{i=1}^{m} w_i \epsilon_i)}$. This can also be understood as an approximation supported by Lindeberg's Central Limit Theorem, which only requires that the random variables being linearly combined have finite variance, satisfy Lindeberg's condition, and be independent. Lindeberg's condition indicates that none of the random variables have a disproportional relevance to the calculation of the variance of the portfolio [85].

This way, the optimization problem to be solved is:

$$\min_{\mathbf{w}} \quad U(w, \epsilon_1, \ldots, \epsilon_m)$$

$$\text{s.t.} \quad \mathbf{w}^T \mathbf{e} = 1 \tag{49}$$

$$\text{s.t.} \quad w_i \geq 0, \forall i \tag{50}$$

where $U(w, \epsilon_1, \ldots, \epsilon_m)$ is a heuristic utility function on the weights and the residues. Several of those utility functions are discussed subsequently.

- MinADWeights

  For this case, the utility function is:

$$U(w, \epsilon_1, \ldots, \epsilon_m) = \frac{2\sigma_{stacked} AD_{stacked}}{\sigma_{stacked} + AD_{stacked}} \tag{51}$$

which is nothing but the harmonic mean of the standard deviation of the portfolio ($\sigma_{stacked}$) and the modified Anderson-Darling statistic [86] calculated for the normalized portfolio residues ($AD_{stacked}$). This modified statistic considers that both mean and variance are unknown. Mathematically, both the mean of the portfolio and its variance are estimated from the sampled residues, $T = \sum_{i=1}^{m} w_i \epsilon_i / \sigma_{stacked}$, $\Phi(z)$ is the cdf of a standard normal random variable and:

$$AD_{stacked} = \left( -n - \frac{1}{n}\sum_{i=1}^{n}(2i-1)(\ln \Phi(T_i) + \ln(1 - \Phi(T_{n+1-i}))) \right)\left( 1 + \frac{4}{n} - \frac{25}{n^2} \right) \tag{52}$$

- NormWeights

  In this case, the utility function is simply $U(w, \epsilon_1, \ldots, \epsilon_m) = \sigma_{stacked}$, where $\sigma_{stacked}$ has been obtained as maximum likelihood estimate for the variance of a Normal random variable with zero-mean fitted to $\sum_{i=1}^{m} w_i \epsilon_i$.

- MaxLWeights

In this case, the utility function is the negative log-likelihood of the samples from $\sum_{i=1}^{m} w_i \in_i$ being distributed as a Normal random variable with zero-mean and standard deviation $\sigma_{stacked}$, which is estimated from the samples of $\sum_{i=1}^{m} w_i \in_i$.

- MinKLWeights

In this case, the utility function is the KL divergence between two normal random variables $P$ and $Q$, where $P$ has mean zero and standard deviation $\sigma_{stacked}$ and $Q$ has mean $\mu_{stacked}$ and standard deviation $\sigma_{stacked}$. All these moments are estimated from the samples of $\sum_{i=1}^{m} w_i \in_i$.

## HPOSS and its application in a study case

A study case shall be conducted to show how the new HPOSS methodology performs compared to other alternatives. Thus, consider the four-variable I-beam problem taken from [87]. The critical response for this problem is the maximum bending stress $\zeta_{max}$ developed in a simply supported beam with 1 m of length after a point load $P$ is applied at its center, which is calculated as:

$$\zeta_{max} = \frac{Pd_1}{4I}; \quad I = \frac{1}{12}\left(d_2 d_1^3 - (d_2 - d_3)(d_1 - 2d_4)^3\right) \tag{53}$$

where each $d_i$ is a dimensional design variable such that $0.1m \leq d_1, d_2 \leq 0.8m$ and $0.009m \leq d_3, d_4 \leq 0.05m$, as specified in [87] and presented in Fig 3.

In the study case, $P = 1000$ N is assumed to be deterministic, and the dimensional variables are all Beta random variables in the design ranges specified. To test the robustness of the techniques involved, five different fundamental beta random variables were chosen, as presented in Fig 4.

For each type of Beta random variable presented in Fig 4, proper linear scaling is performed to adjust the support from (0, 1) to the corresponding physical limits mentioned. All these types were used to demonstrate how the stacking alternatives behave when the sampled points are mainly located to the left (Beta(2,5)), the center (Beta(2,2)), the right (Beta(5,2)) and both ends (Beta(0.5,0.5)) of the domain as well as uniformly distributed over it (Beta(1,1)).

To calibrate the surrogate models, a small dataset of 80 samples of $\zeta_{max}$ is obtained by using the Latin Hypercube sampling algorithm for the input variables, which was implemented using the *chaospy* python package [70]. The DoE was generated as if the input random variables were Uniform in their respective ranges, mainly to provide a space-filling dataset. All sub-regions of the random variable support are represented in the datasets. The dataset can be found in [88], where a .h5 file is present. The file contains a dataset called "Calibrations_LHS", which consists of a $80 \times 5$ numpy array of float numbers corresponding to $d_1(m)$, $d_2(m)$, $d_3(m)$, $d_4(m)$ and $\zeta_{max}(Pa)$, where $d_i$, $i = 1, \ldots, 4$ are dimensions (in meters) of the I-beam and $\zeta_{max}$ is the maximum bending stress (in Pascals) developed in such beam.

The DoE size of 80 samples was chosen to represent a small sample problem, where obtaining newer samples of the unknown response function is unfeasible. Increasing the sample size would not impact the general results and discussions. On the other hand, it is possible that the selected models would be different, which is expected because the behavior captured with more data is different (one gets to see more of the unknown function). This way, the DoE size is just a pre-definition that does not impact on the application of the methodology hereby proposed.

**Model calibration.** From all the possible regressors in *scikit-learn* [66], the ones cited in the Materials and Methods section were considered. As discussed, the algorithm

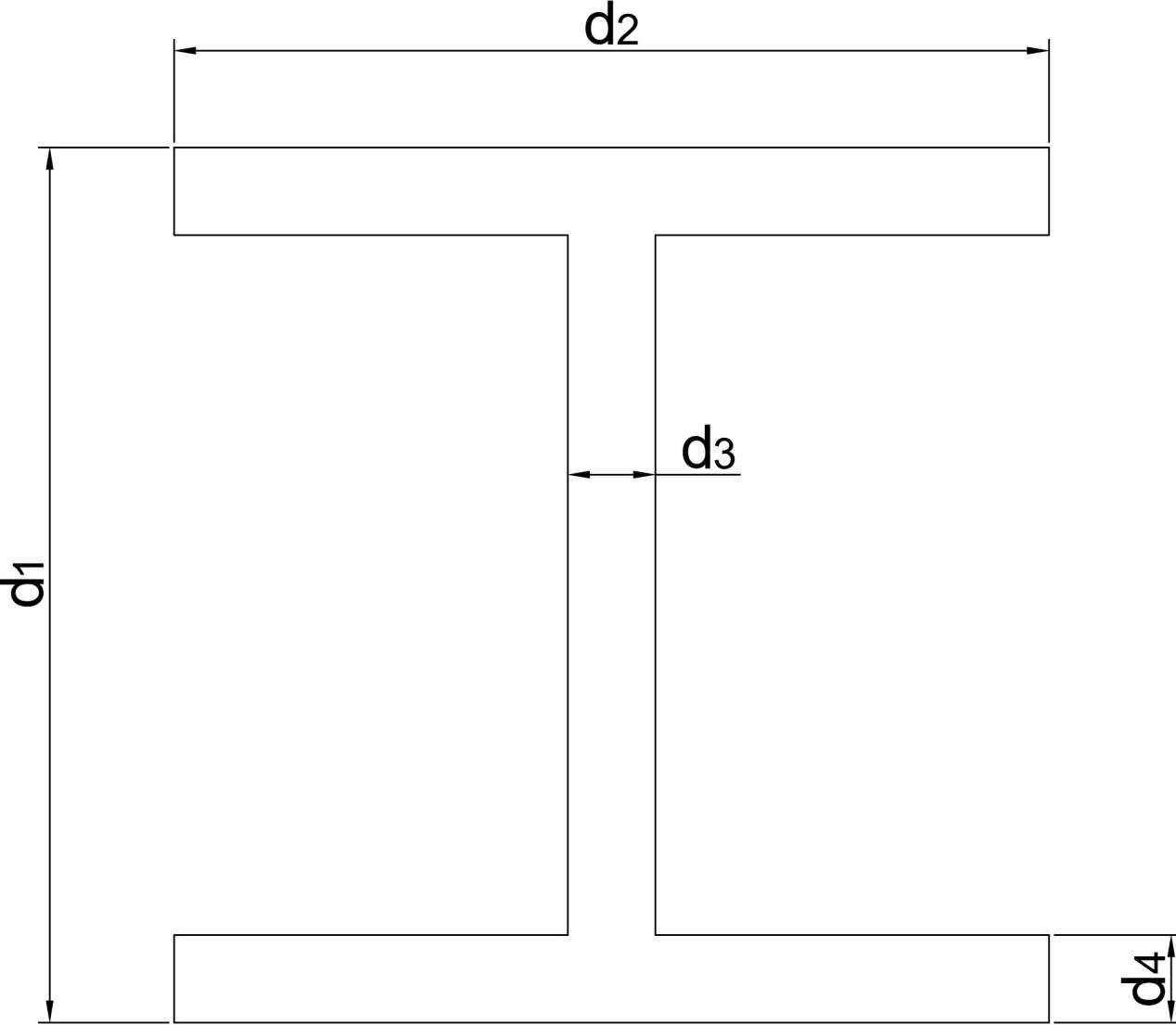

**Fig 3. Dimensional variables of I-beam [87].**

'GaussianProcessRegressor' is a kriging implementation, and no manual hyperparameter tuning was performed for any of the algorithms.

Polynomial Chaos Expansions were fitted to the data using the *chaospy* [70] algorithm. In this case, a point collocation fitting approach was considered. Seven types of multivariate linear regression models, available in *scikit-learn*, were used to obtain the polynomial coefficients to solve the least-squares problem (find the coefficients of the PCE), namely: "least squares", "elastic net", "lasso", "lasso lars", "lars","orthogonal matching pursuit" and "ridge". The maximum order of the polynomials considered was such that the number of unknown coefficients did not exceed the number of training samples. In the case with four input variables and 80 training samples, at most, polynomials of order 4 were fitted. Besides, three base random variables were selected for the expansions: Uniforms in the range (−1, 1), standard Normals, and the Uniforms in the physical ranges mentioned (hereby referred to as Real expansion, in the

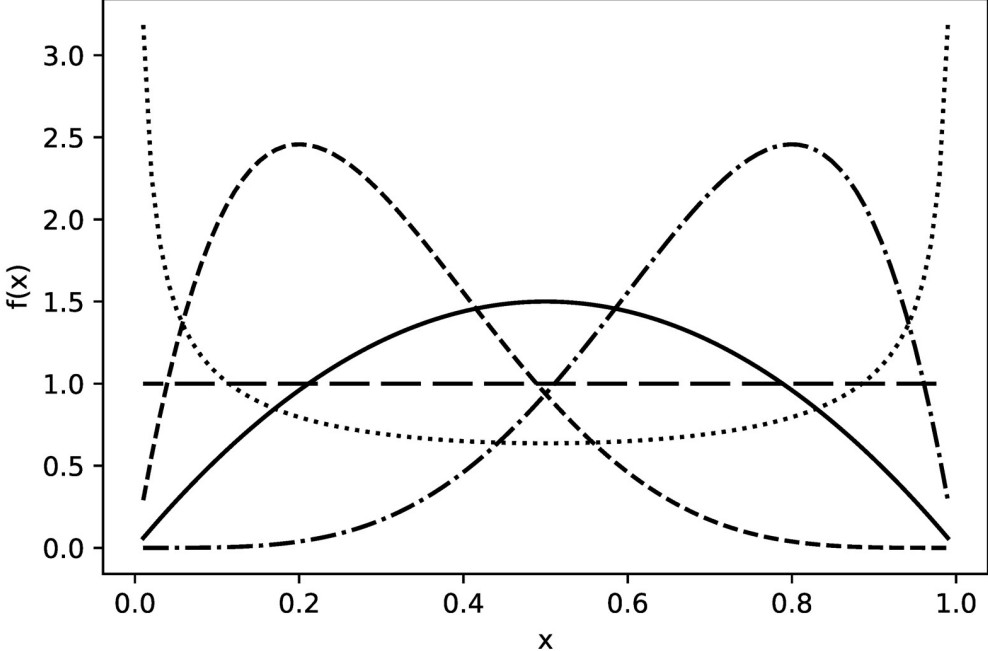

**Fig 4. Five types of Beta functions considered.**

sense of the real random variable). For all the three types of variables, the three terms recurrence method implemented in *chaospy* was used. It is worth noticing that classical Legendre and Hermite polynomials were retrieved for Uniforms in the range (−1, 1) and standard Normals.

In all the cases, the naming convention for the PCE expansions was: *PCE − XXX YYY − Z*, where *XXX* denotes which type of random variable was considered (Unif = Uniform (−1, 1), Norm = standard Normal and Real = shifted Uniforms in the physical ranges); *YYY* denotes which algorithm was used to perform the multi-linear least squares regression and *Z* is the polynomial expansion order.

For the machine learning models, it is important to highlight that scaling of the input values was performed by using the function *MinMaxScaler* of the *scikit-learn* package, which scales and translates each feature individually such that it is within a given range on the training set (in our case, between zero and one).

Overall, a total of 124 models were trained using the dataset described: 40 Machine Learning models and 84 PCE-based models (3 types of random variables considered fitted using 7 least square algorithms and up to fourth order expansions, i.e., $3 \times 7 \times 4 = 84$). Among those, 99 models could be fitted without convergence and other numerical issues.

**Model selection.** For the DBHT algorithm, the similarity measure chosen was the generalization error similarity. The clusters obtained from running the DBHT algorithm with different distance metrics (Pearson, Kendall' $\tau$ and Spearman's $\rho$ correlation distances, approximate ratio correlation distance and approximate normal correlation distance) were also evaluated, and the clusters obtained for the Pearson, Kendall' $\tau$ and Spearman's $\rho$ correlation distances

are presented in Figs 5–7. The "PCE Real Lasso Lars-4" model is highlighted since this would be chosen based on the minimum observed generalization error. The selected cluster is the one that contains this model. It is important to highlight that the optimal number of clusters was established by using the two-order difference to gap statistic [89] implemented in the python package *Riskfolio-Lib* [74].

The analysis of Figs 5–7 indicate that the DBHT algorithm with both Kendall' $\tau$ and Spearman's $\rho$ correlation distance metrics provided the same clustering result (same models clustered together with the "PCE Real Lasso Lars-4" model). Moreover, with the use of Pearson correlation distance metric, the DBHT algorithm provided a bigger cluster which contains not only the models clustered by DBHT using the other distance metrics, but also extra models. This indicates that if the Pearson correlation is used, a looser clustering process is carried out, which ends up including more models in the final cluster.

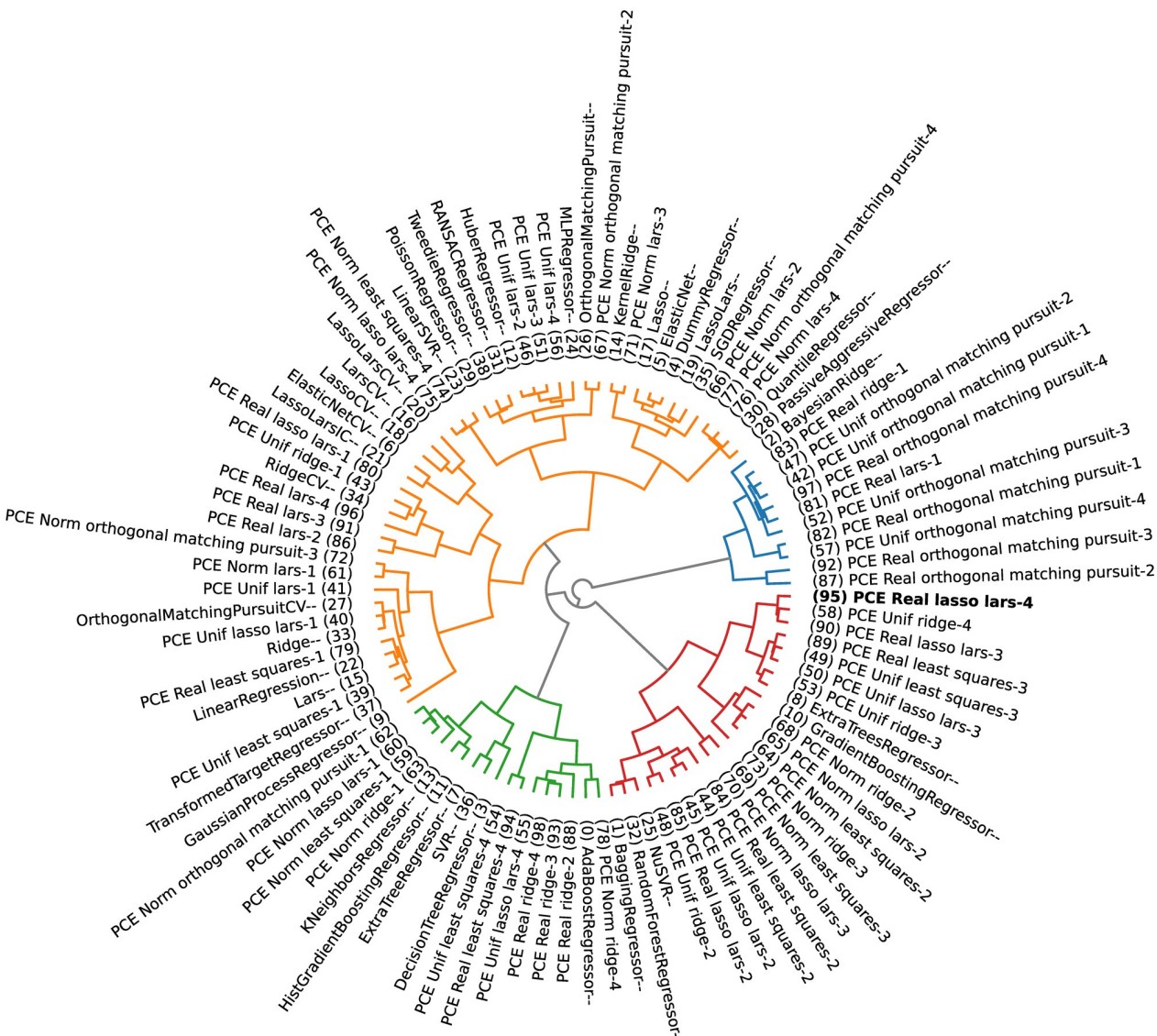

**Fig 5. Radial dendrogram obtained from DBHT algorithm with Pearson correlation distance and generalization error similarity.**

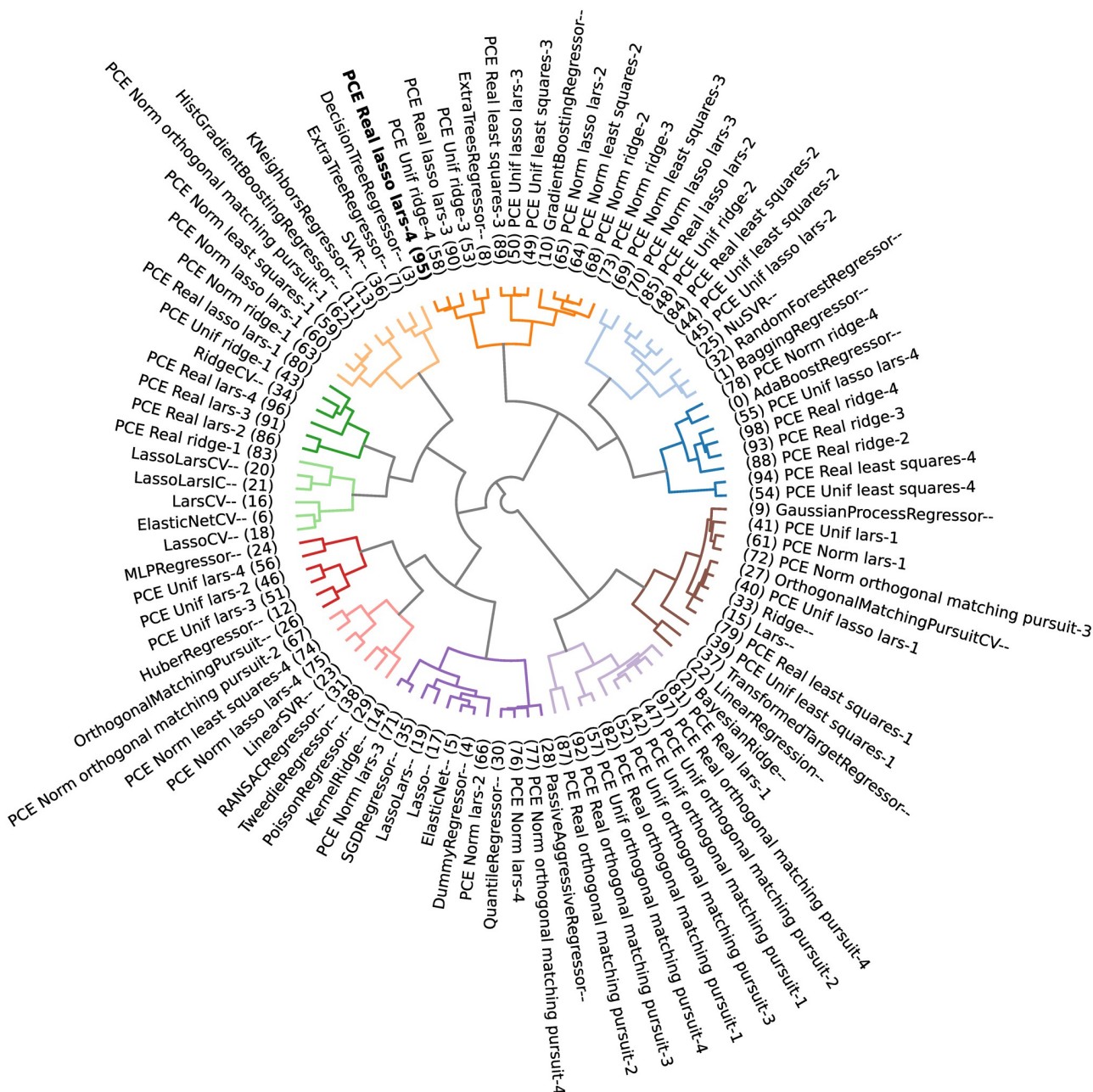

**Fig 6. Radial dendrogram obtained from DBHT algorithm with Kendall' $\tau$ correlation distance and generalization error similarity.**

In the present paper, we seek to use the DBHT algorithm to filter among a large number of candidate models. Therefore, while we intend to eliminate a good share of uncorrelated models, we still want to have a good number of algorithms in the final cluster. Therefore, we chose the Pearson correlation distance metric to carry out the study case. Depending on the specific application envisioned, the reader may, on the other hand, choose a tighter clustering scheme, by selecting either Kendall' $\tau$ or Spearman's $\rho$ correlation distance metrics together with the DBHT algorithm.

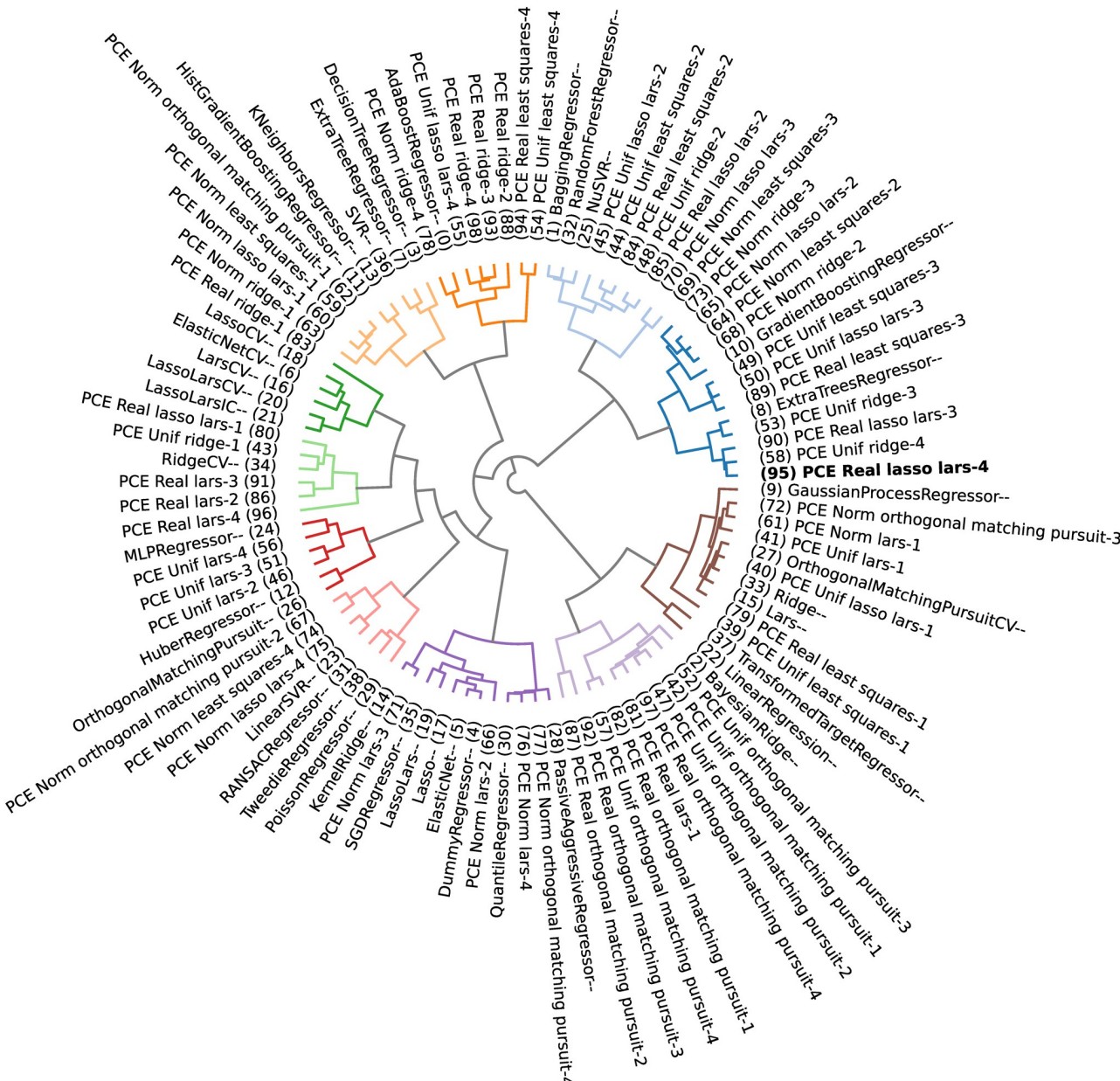

**Fig 7. Radial dendrogram obtained from DBHT algorithm with Spearman's $\rho$ correlation distance and generalization error similarity.**

By analyzing Fig 5, the combination of the Pearson correlation distance with the generalization error similarity provides a good combination of low generalization errors and highly correlated residue random variables. This can be visualized by noticing that, on average, the bubbles formed by the DBHT algorithm gather low generalization error models (similarity defines the bubbles and we chose the cluster which contains the model with minimum generalization error) and the internal hierarchy is defined by models with similar analytical capabilities (highly correlated leave-one-out residues). Similar analytical capabilities can be confirmed by the fact that, in general, the clusters obtained gathered similar types of PCEs and linear regression methods.

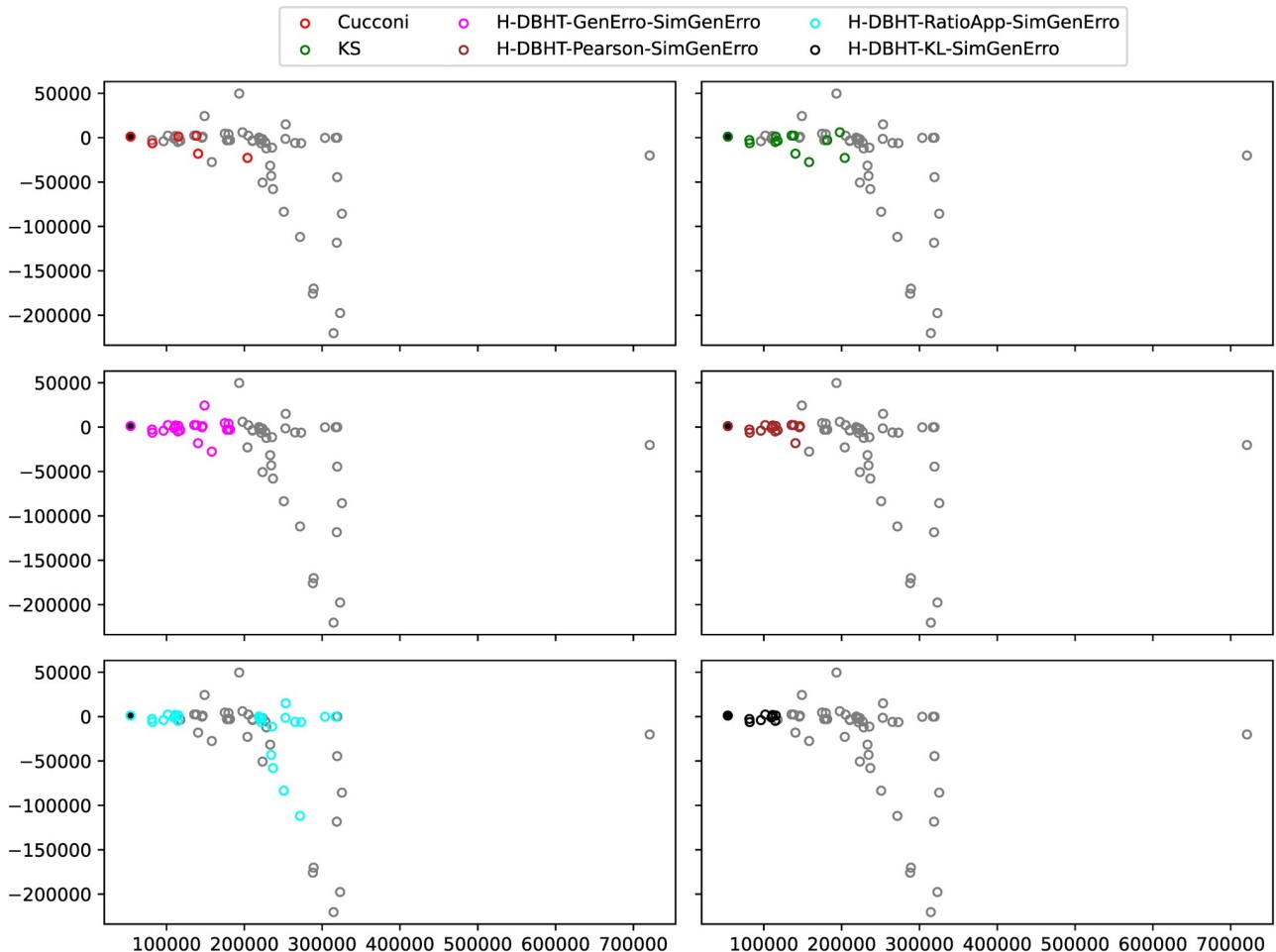

**Fig 8. Model selection for each clustering algorithm considered.** For color blindness accessibility, the legend's first(second) row describes the first (second) column of the grid.

Fig 8 presents the selected models in the bias-square root of variance space to better visualize how each combination of metrics leads to different clustering scenarios. The full black dot in Fig 8 represents the "PCE Real Lasso Lars-4" model.

The legend in Fig 8 represents the hypothesis test approaches (Cucconi and KS) as well as the DBHT algorithm with the generalization error similarity (SimGenErro) and different distance metrics: generalization error distance (GenErro), Pearson correlation distance (Pearson), approximate ratio distance (RatioApp) and Normal approximation ratio distance (KL).

By considering the DBHT algorithm with Pearson correlation distance and generalization error similarity, from the 99 fitted models, a subgroup of 23 was selected to be stacked. This represents a reduction of about 77% in the total number of models, representing a good filtering scheme which still preserves diversity.

**Weights calculation.**   A total of eleven methodologies were considered to calculate the stacking weights. All the optimization procedures were solved using the Sequential Least Squares Programming (SLSQP) algorithm of *scipy* [84]. Fig 9 presents the bounded problems' resulting weights. For unbounded procedures, the weights ranged from -30000 to 30000 and are not represented in Fig 9 due to scale issues.

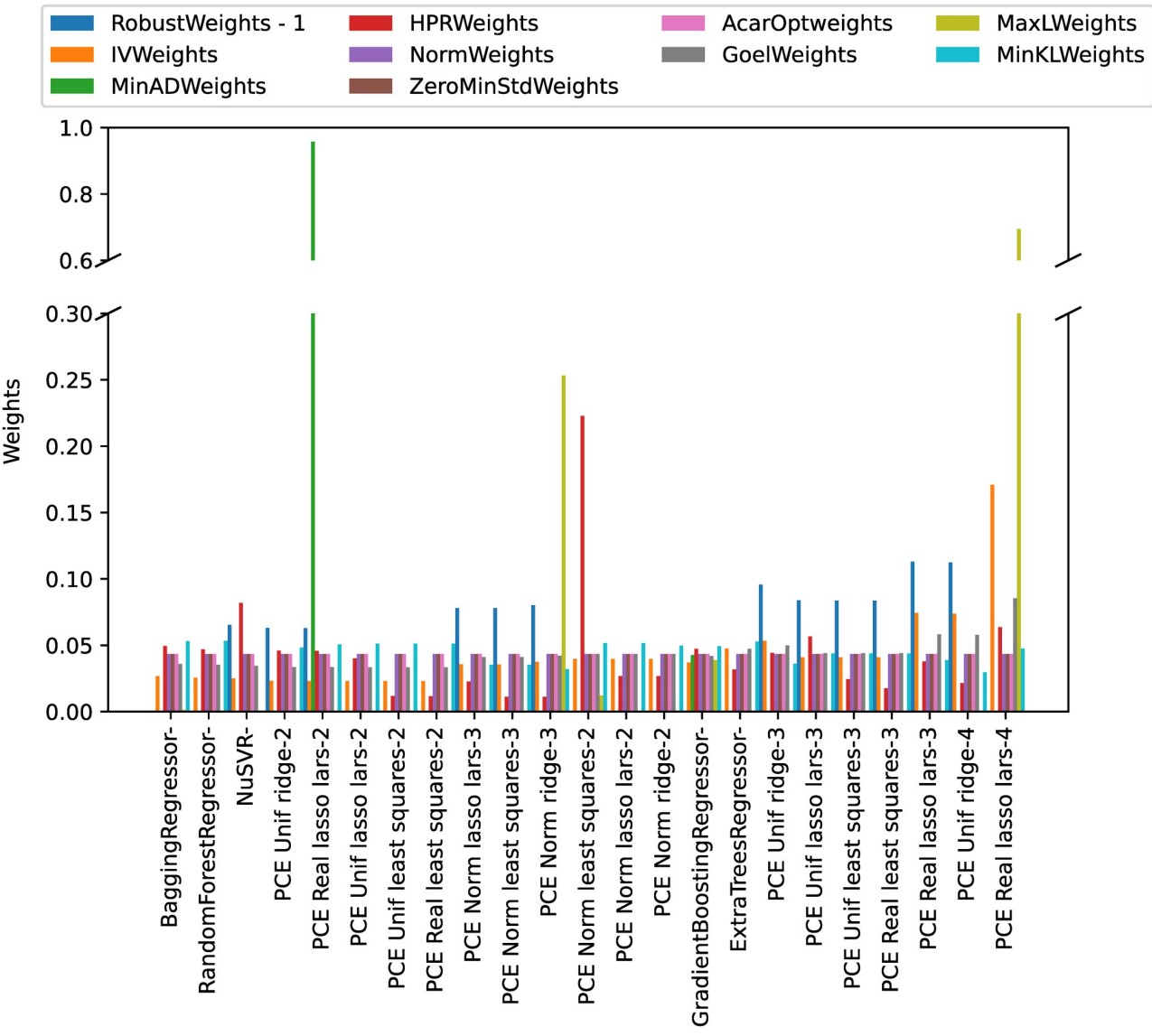

**Fig 9. Results for all bounded weights stacking strategies.**

The analysis of Fig 9 reveals that except from the "MinADWeights", "MaxLWeights", "HPRWeights", "IVWeights" and "RobustWeights—1", all the other methodologies tended to assign comparable weights to all the candidate models.

In special, Fig 9 shows that "MinADWeights" present an unbalanced scheme where too much importance was given to the "PCE Real lasso lars-2" model. This illustrates how a pure optimization approach may lead to less interpretable weights. Fig 10, on the other hand, presents a heatmap for the methods whose maximum weights were lower than 0.3.

Figs 9 and 10 also indicate that the novel heuristic methods "NormWeights", "ZeroMinStd-Weights", and "AcarOptWeights" all resulted into the simplest weighting scheme: equal weights for all the models. This indicates that the convergence of the optimization problem was either problematic (since the initial weights used as first guesses were precisely equal ones) or that the equal-weight portfolio is the optimal one. Optimizing the "AcarOptWeights" by

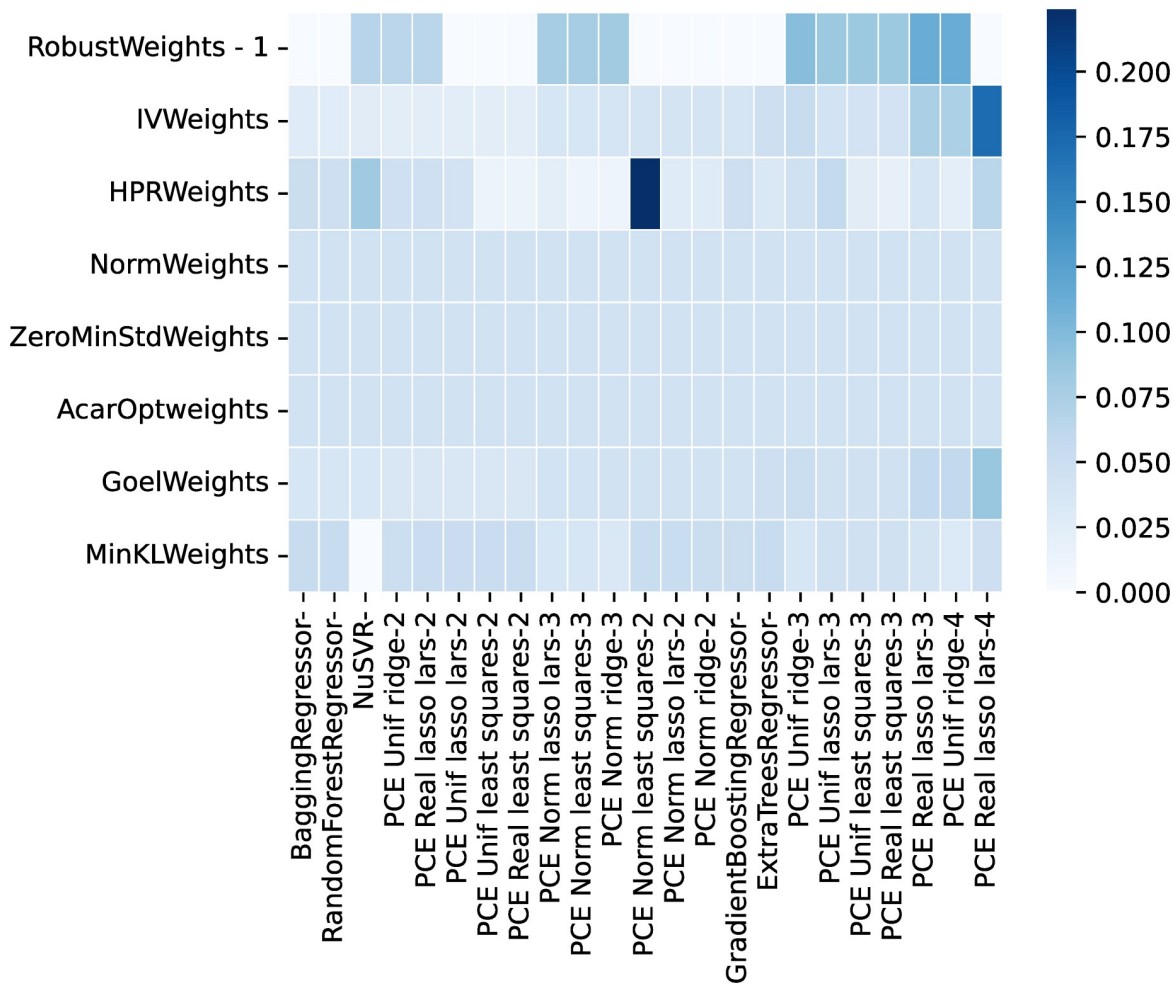

**Fig 10. Heatmap of weights.**

coupling it to the CLA algorithm of Markowitz did not provide convergent results when using the CLA algorithm presented in [45]. It is possible that the highly correlated and, sometimes, linearly dependent $\epsilon_i$ prevented convergence. Overall, optimization approaches struggled to converge to meaningful results. We chose to keep such results to illustrate how pure optimization techniques can be problematic and to indicate that direct approaches, such as the HPR strategy, are of great interest in this matter, as the "optimal" solution is deterministic and explicit. In other words, the HPR strategy can always provide weights, regardless of a possible ill-conditioning of the covariance matrix.

On the other hand, "GoelWeights", "IVWeights" and "MaxLWeights" gave too much credit to the model with the least observed generalization error. Also, the "RobustWeights—1" provided an interesting result, where several models were almost discarded from the analysis as low weights components. The type of robust optimization hereby considered privileges lower bias at the cost of higher variance. Besides, the "HPRWeights" were such that two ordinary algorithms ("NuSVR" and "PCE Norm least squares—2") were given considerably larger weights, while some other algorithms were almost disregarded with comparably lower weights.

**Probability of failure calculations.** The plots in Figs 11–15 present the prediction intervals for the probabilities of failure calculated for each type of input random variable and the corresponding performance of the individual algorithms and stacking strategies. The prediction interval is defined by its boundaries $[p_{int,upper}, p_{int,lower}]$ and was obtained as the 95% bias-corrected and accelerated bootstrap confidence interval (BCa) of the mean probability of failure after a bootstrapping prediction assessment for 300 times. This number of bootstrap realizations was considered adequate after numerical experiments. The bootstrap confidence intervals for the prediction values were obtained by applying the *bootstrap* method from *scipy.stats* Python package [84].

To properly compare the plots, it was considered that a failure happens whenever $\zeta_{max} > \zeta_{0.1} = 786260.04$ Pa, where $P(\iota > \zeta_{0.1}) = 0.1$ and $\iota$ is the random realization of the exact model.

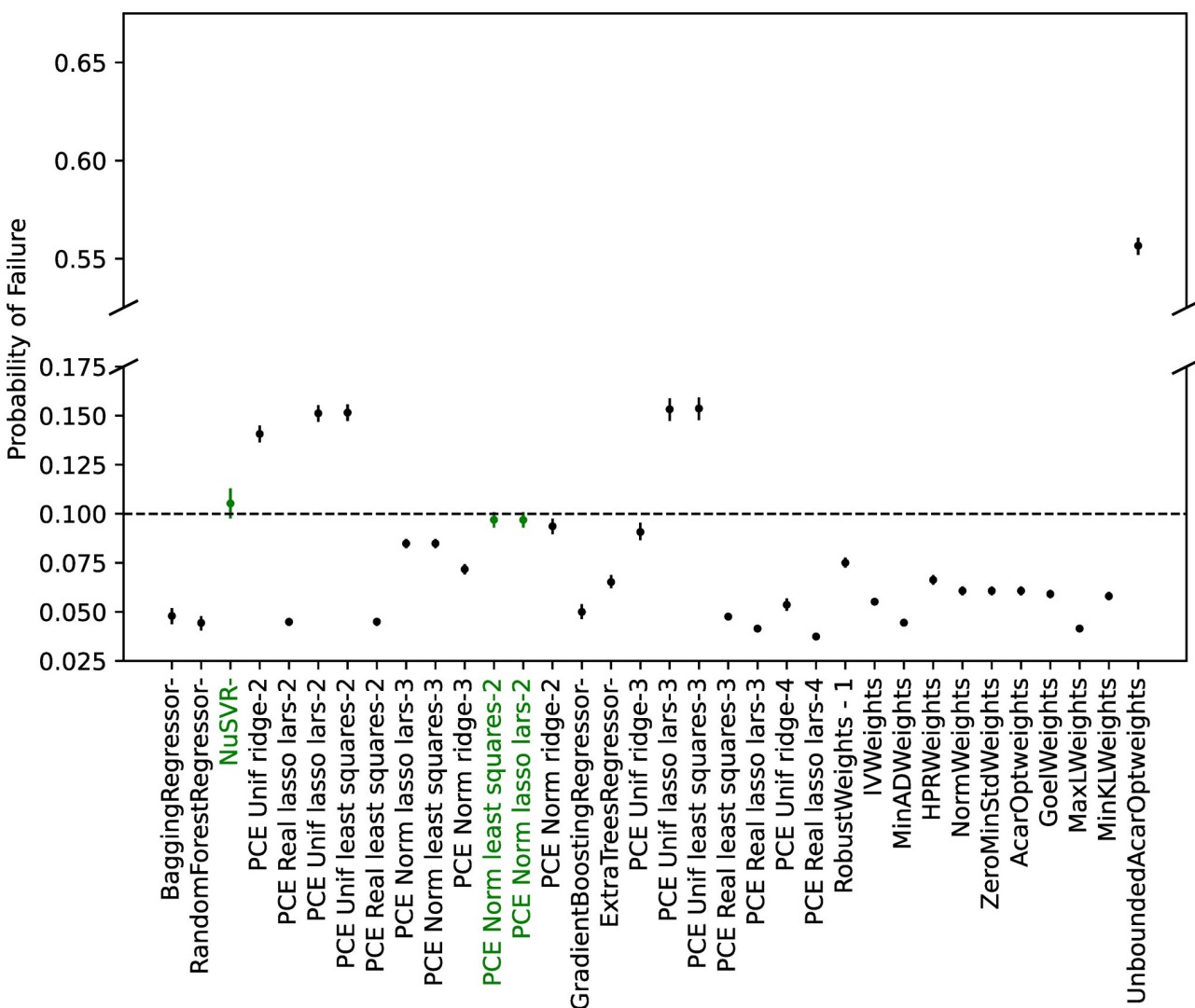

**Fig 11. Prediction intervals of probabilities of failure considering that the input variables are linearly scaled Beta(2,5), i.e., the center of mass of the distribution closer to the left end of the interval, according to their respective physical ranges.**

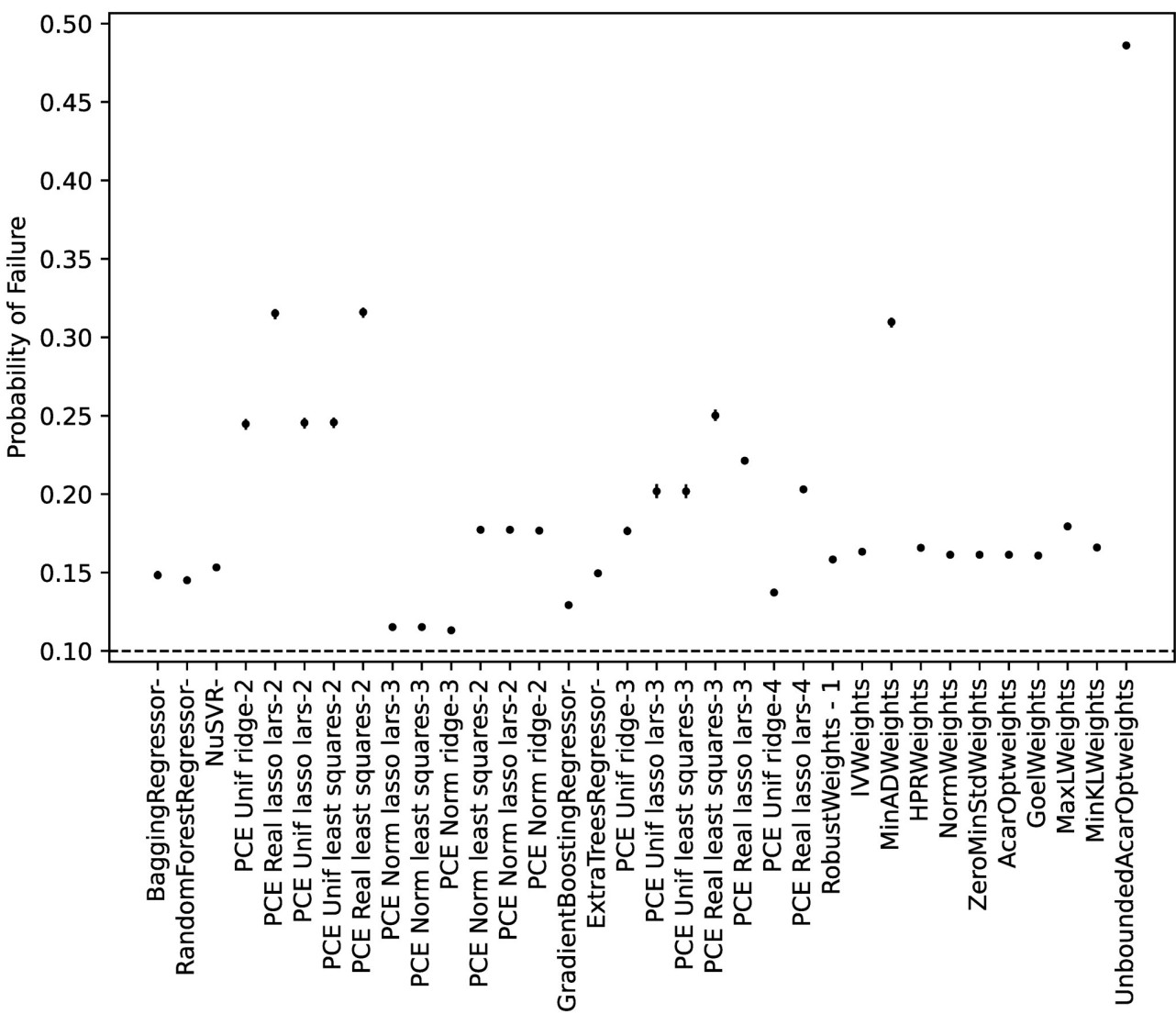

**Fig 12. Prediction intervals of probabilities of failure considering that the input variables are linearly scaled Beta(2,2), i.e., the center of mass of the distribution at the center of the interval, according to their respective physical ranges.**

Also, whenever a model or stacking strategy provided a prediction interval that contained the correct value (which was chosen as 0.1), it was colored green.

To assess how each algorithm and staking strategy performed, the following metric was defined to quantify how far the true probability of failure value was from the confidence interval. Thus, let $v$ be the performance metric defined as:

$$v = v_a + v_b, \tag{54}$$

where $v_a = 0$ if the prediction interval contains the correct probability of failure and $v_a = \min(|p_{int,upper} - 0.1|, |p_{int,lower} - 0.1|)$ otherwise. Also, $v_b = p_{int,upper} - p_{int,lower}$.

In other words, $v_a$ quantifies how far the actual probability of failure is from the predicted interval by either checking if this value is inside the gap (thus $v_a = 0$) or which is the smallest distance from the correct value to the boundaries of the confidence interval. Also, $v_b$ quantifies

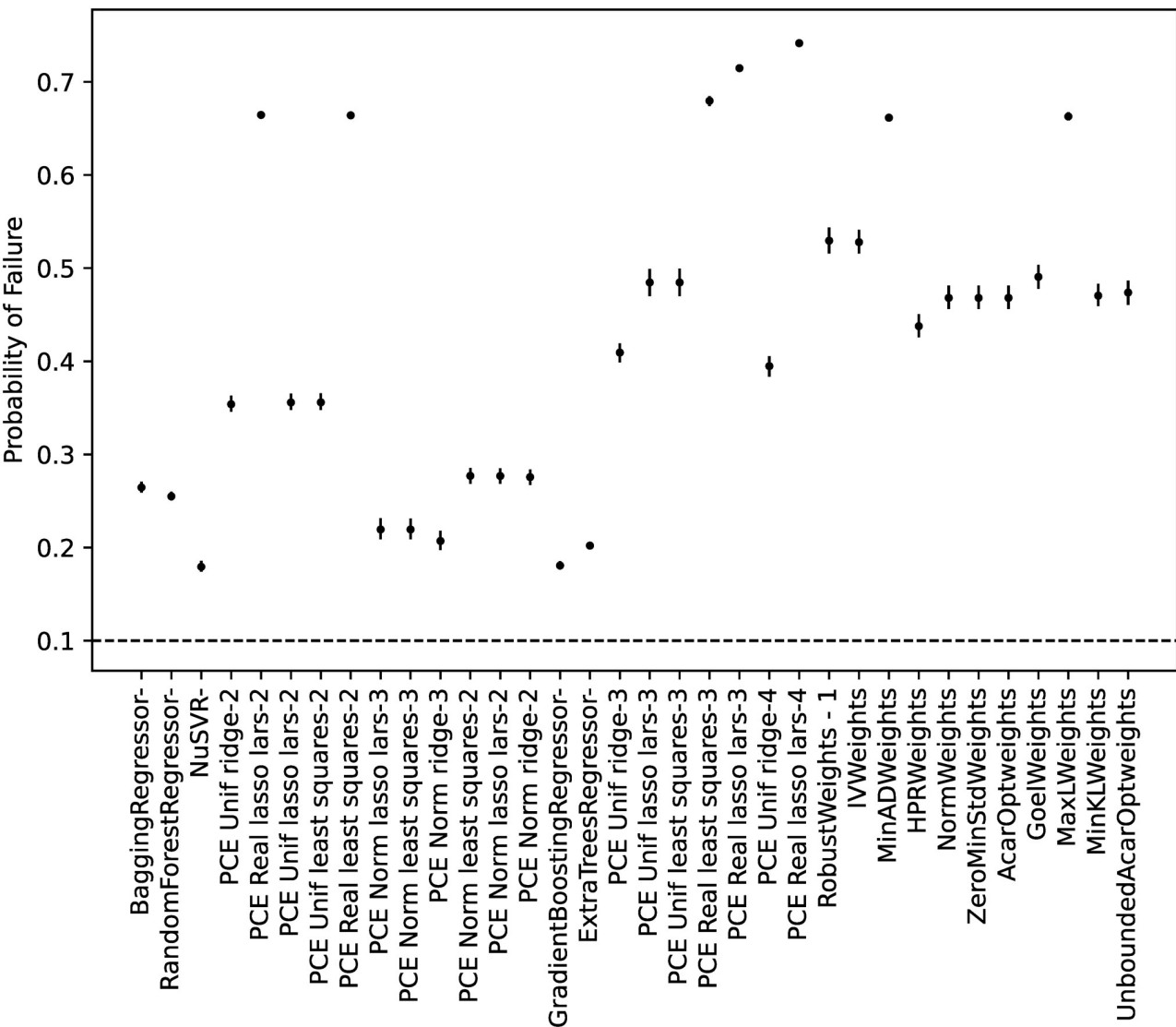

**Fig 13. Prediction intervals of probabilities of failure considering that the input variables are linearly scaled Beta(5,2), i.e., the center of mass of the distribution closer to the right end of the interval, according to their respective physical ranges.**

how wide is the confidence interval. Overall, the best models would provide prediction intervals that contained the actual value, and the gap itself should be narrow such that lower values of $v$ indicate better model performance.

Fig 16 shows that the HPR stacking strategy has the lowest mean performance metric among all the stacking strategies compared. However, this indicates that the HPR strategy performs well even when the random input variables have completely unseen probability distributions, indicating its performance can be considered superior to other methods.

In general, the lower the value of $v$, the better the model. On the other hand, stacking is the best possible model choice, not necessarily the result with the lowest error metric. This comes from the fact that, usually, it is not possible to assess which is the individual model which presents the lowest metric (in our example, the PCE Norm ridge-3) because we do not know the function we are trying to approximate but only a few samples from its evaluation. Therefore,

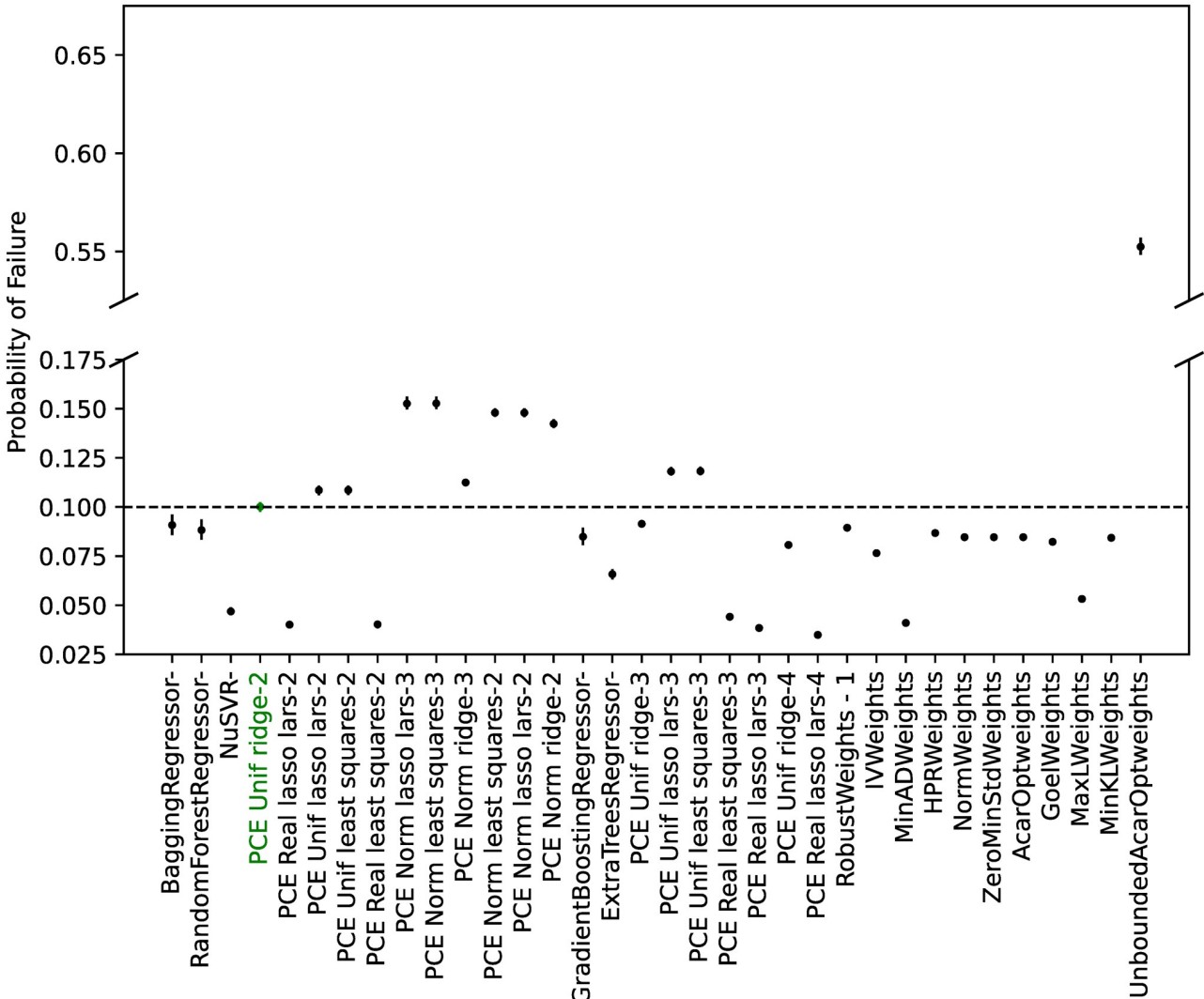

**Fig 14. Prediction intervals of probabilities of failure considering that the input variables are linearly scaled Beta(0.5,0.5), i.e., two centers of mass for the distribution located at both the right and left ends of the interval, according to their respective physical ranges.**

stacking becomes the best possible choice to provide an estimate in this context, as we don't know how well each model will mimic the objective function. Besides, it can be seen that the best overall model (which has the lowest error metric) does not even have the lowest LOO generalization error. There is, likewise, no hint that allows us to choose it. In other words, it is impossible to pin down the best model beforehand, so stacking is necessary.

The study case carried out revealed some interesting aspects of the stacking problem, which follow:

- Intervals lengths:

Consistently, the prediction intervals are narrower for an ensemble of models than for single models. This is a known and expected result, as stacking models can be viewed as a variance reduction technique.

- PCE expansion quality:

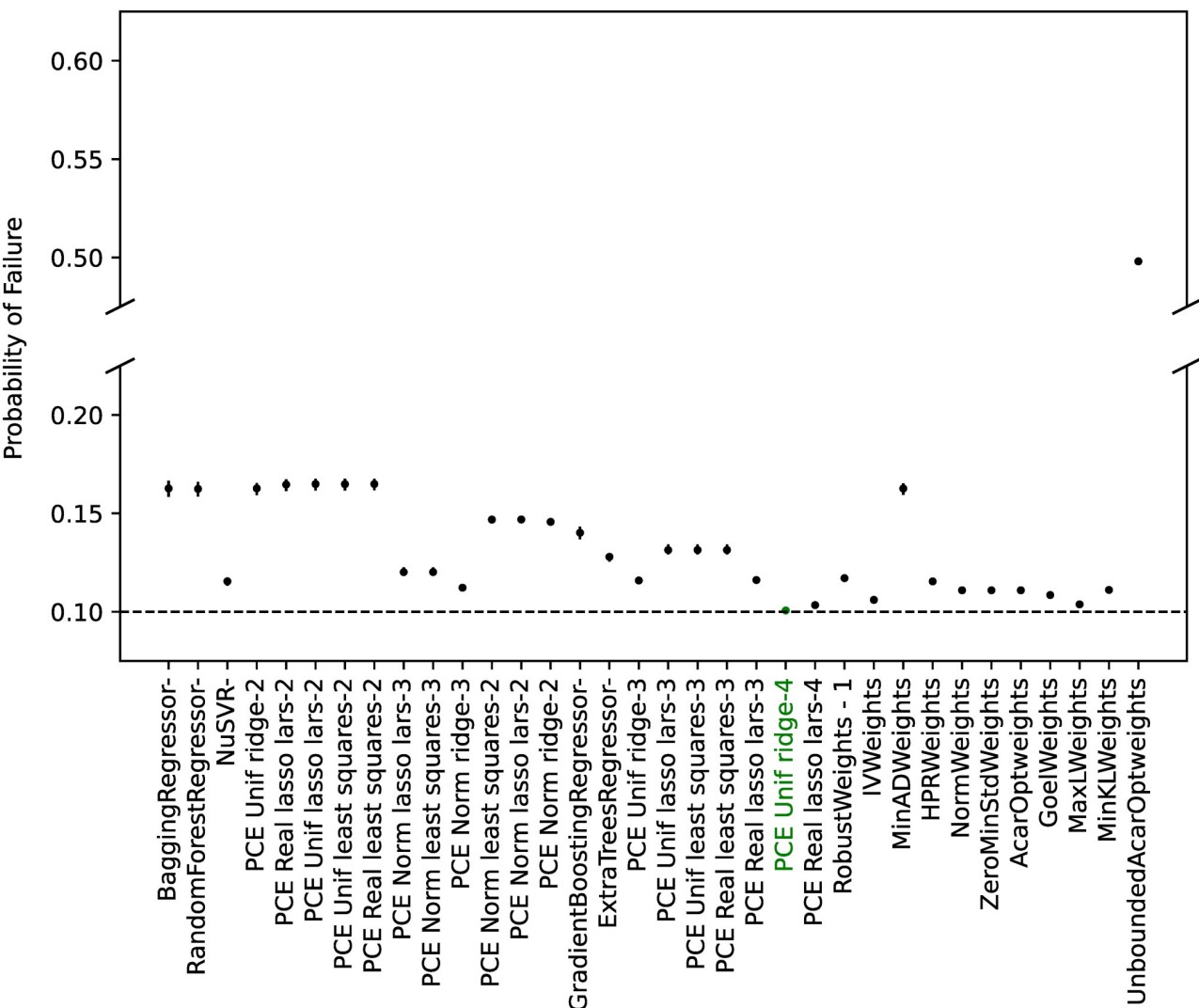

**Fig 15. Prediction intervals of probabilities of failure considering that the input variables are linearly scaled Beta(1,1), i.e., uniform distribution over the interval, according to their respective physical ranges.**

Overall, custom PCE expansions for the same input random variables (even if they are slight modifications of traditional random variables with known closed-form PCEs) consistently perform worse than the conventional expansions (Uniforms, Normals).

Stacking strategies that ended up assigning higher weights to these custom ("Real") expansion models got a severe performance hit. Also, except for the "MinADWeights" ensemble method, all individual custom ("Real") expansions performed worse than the ensemble of models.

- Overall performance of surrogate models:

Very few algorithms provided prediction intervals that contained the actual probability of failure value. If a winner-takes-all approach had been considered, the algorithm with the lowest observed generalization error (PCE-Real lasso lars-4) would have been selected. On the other hand, such a model had a poor out-of-sample performance overall, which highlights why using ensembles is a good idea.

- Overall performance of stacking strategies:

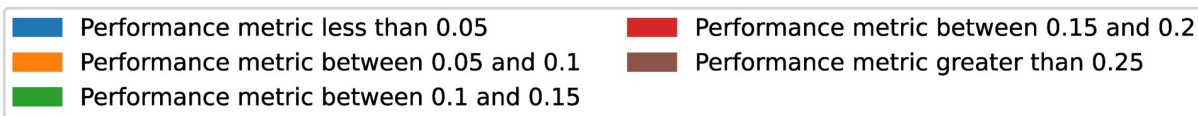

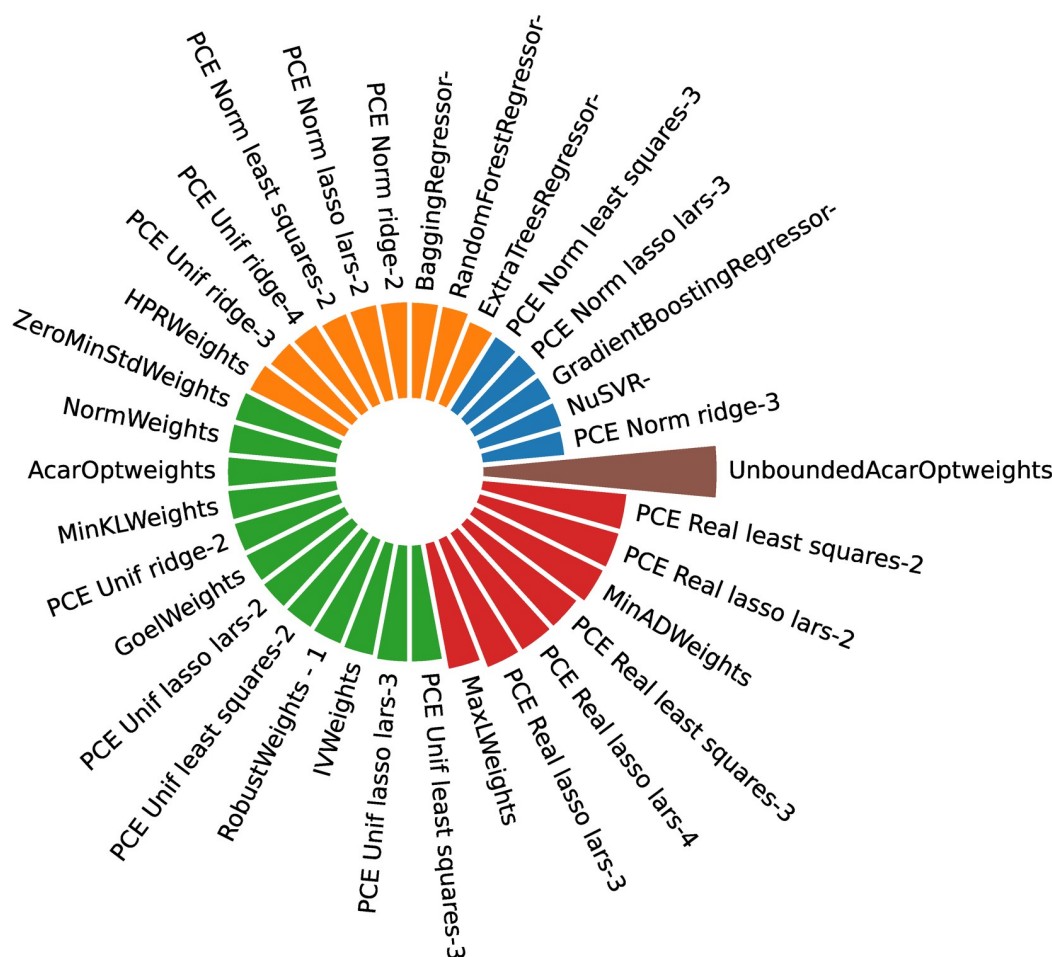

**Fig 16. Comparison of mean performance metric for probabilities of failure calculations considering all the input variables.**

Traditional optimization techniques, such as the unconstrained one by Acar & Rais-Rohani [19], did not perform well. The unbounded nature of the weights resulted in a volatile weighting scheme. Even the constrained version and the heuristic approach proposed by Goel et al. [18] did not outperform the proposed HPR alternative. Overall, stacking choices that involved optimization either did not converge or converged to a meaningless set of weights due to the correlation and, sometimes, linear dependence between the $\epsilon_i$ random variables samples.

## Conclusions

The present paper proposes a novel stacking strategy for surrogate models. Reinterpreting stacking problems as portfolio management and optimization situations allows several alternatives to combine individual models better.

A two-step methodology is proposed: first, models are calibrated, and, based on their leave-one-out residues, a subset of surrogates is chosen to be stacked. To illustrate the application of the methodology, a study case was performed and revealed that, among 99 fitted models, the Directed Bubble Hierarchical Tree—DBHT—algorithm (with a generalization error similarity matrix and a Pearson correlation distance matrix) was able to cluster a small subgroup of only 23 models that were worth staking. This represents a reduction of about 77% in the total number of models, representing a good filtering scheme which still preserved analytical diversity.

By considering a probability of failure example, a custom metric was defined to quantify how far the true probability of failure value was from the confidence interval provided by each stacking alternative. This metric revealed that the best linear weighting scheme was the Hierarchical Risk Parity method (HPR), despite the relative quality of the other stacking strategies proposed in the present paper.

The "RobustWeights—1" method could benefit from changing the uncertainty set definition and distribution. However, the regularization characteristics of this type of approach seem promising and worth studying in subsequent papers.

HPR algorithm tends to balance weights according to how similar the analytical structure of the methods is. In this regard, the hierarchical nature of the process assures that surrogates in the same hierarchical branch receive equal weights, increasing model diversity and avoiding assigning higher weights to a single specific class of surrogates.

## Acknowledgments

All the authors acknowledge the support of the Post-graduate program in aeronautical infrastructure engineering, Aeronautics Institute of Technology (ITA) for hosting L.C.S.M.O. and V.R.D. as postdoctoral researchers.

## Author Contributions

**Conceptualization:** Luan Carlos de Sena Monteiro Ozelim, Vinicius Resende Domingues.

**Formal analysis:** Luan Carlos de Sena Monteiro Ozelim, Dimas Betioli Ribeiro.

**Funding acquisition:** Dimas Betioli Ribeiro.

**Investigation:** Luan Carlos de Sena Monteiro Ozelim, Dimas Betioli Ribeiro, Vinicius Resende Domingues, Paulo Ivo Braga de Queiroz.

**Methodology:** Luan Carlos de Sena Monteiro Ozelim.

**Software:** Luan Carlos de Sena Monteiro Ozelim.

**Supervision:** Dimas Betioli Ribeiro, José Antonio Schiavon, Paulo Ivo Braga de Queiroz.

**Writing – original draft:** Luan Carlos de Sena Monteiro Ozelim.

**Writing – review & editing:** Dimas Betioli Ribeiro, José Antonio Schiavon, Paulo Ivo Braga de Queiroz.

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
