## [Decision Letter · Decision Letter 0]

20 Jun 2023

PONE-D-23-08481HPOSS: A hierarchical portfolio optimization stacking strategy to reduce the generalization error of ensembles of modelsPLOS ONE

Dear Dr. Ozelim,

Thank you for submitting your manuscript to PLOS ONE. After careful consideration, we feel that it has merit but does not fully meet PLOS ONE’s publication criteria as it currently stands. Therefore, we invite you to submit a revised version of the manuscript that addresses the points raised during the review process.

Please address both reviewer's comments, although I would add certain provisos:-regarding the listing of scikit-learn library (p6) is not completely  unnecessary but certainly useless for the reader the way it is. It should be more instructive to mention the model by classes (linear regressions all available, trees models, neural networks, etc) and which constitute emsembles. This list can then be shorten or substitute by a summary.- describe the parameters considered for tuning- provide the data (one reviewer is not clear about it , and it is a requirement of PLOSONE to have data)- Distance metrics among assets  (p 10) it is implied this is amomg the financial time series of prices or returns, for which I am not particularly fond of Pearson distance metric as it assumes a linear relation among the variables. I'll rather recommend instead Spearman or Kendall which captures comovements or much better the distance correlation, which characterizes independence. In the realm of clustering time series these latter are more effective than Pearson. See for example a study I did on this some time ago: A. Arratia & Marti Renedo (2016). Clustering of exchange rates and their dynamics under different dependence measures (PDF). In: ECML-PKDD. Proc. 1st Workshop MIDAS (MIning DAta for financial applicationS). September 19, 2016 - Riva del Garda, Italy. CEUR-WS, v. 1774, pp 17-28.https://ceur-ws.org/Vol-1774/- Regarding robust optimization (Eq 35) I wonder if one can simplify the maxmin problem to one of maximization:in vectorize notation ( with d=delta):  x(p+ dz) = xp + xdz >= xp - ||xd||||z|| (by Cauchy-Schwarz)   >= xp - ||xd||  , since ||z|| < 1hence the problem becomesmaximize_x: xp - ||xd|| ,  subject to  sum x_i = 1, x > 0so there is no need to sample random variable Z... is this correct?

We look forward to receiving your revised manuscript.

Kind regards,

Argimiro Arratia, Ph.D.

Academic Editor

PLOS ONE

“All the authors acknowledge the support of the Post-graduate program in aeronautical

infrastructure engineering, Aeronautics Institute of Technology (ITA) for providing the

grants to pay the APCs and for providing postdoctoral fellowships to L.C.S.M.O. and

V.R.D..”

“The authors received grants to cover the APC costs from the Coordination for the Improvement of Higher Education Personnel (CAPES) - Programa de Apoio à Pós-Graduação (PROAP) .The funders had no role in study design, data collection and analysis, decision to publish, or preparation of the manuscript.”

Reviewers' comments:

Reviewer's Responses to Questions

**Comments to the Author**

1. Is the manuscript technically sound, and do the data support the conclusions?

Reviewer #1: Yes

Reviewer #2: Yes

2. Has the statistical analysis been performed appropriately and rigorously? 

Reviewer #1: Yes

Reviewer #2: Yes

3. Have the authors made all data underlying the findings in their manuscript fully available?

Reviewer #1: No

Reviewer #2: Yes

4. Is the manuscript presented in an intelligible fashion and written in standard English?

Reviewer #1: Yes

Reviewer #2: Yes

5. Review Comments to the Author

Reviewer #1: The evaluation and construction methodology for a bagging (or ensemble) method presented in the paper is interesting and innovative, making it suitable for submission. The paper covers various aspects of machine learning concepts, such as estimating generalization error through cross-validation, as well as structural reliability concepts like the definition of limit state functions. It makes the paper very complete and at the same time not especially clear.

For PLOS ONE, I recommend a major revision, not in terms of content, but to enhance the overall clarity of the message.

Here are some suggestions for the author prior to publication:

- Consider different assumptions regarding the readers' existing knowledge.

- Detailed information about the scikit-learn library (p6) is unnecessary.

- Explanations of the loss function, limit-state function, polynomial chaos expansion, and Kolmogorov test are not crucial to the core of the paper, as they are well-known in various communities. References are provided for interested readers.

- On the other hand, calibration, being a fundamental step in the proposed procedure, could benefit from more detailed explanations, as it may vary across different communities such as machine learning, deep learning, and reliability.

- Include explicit results of the proposed approach at multiple stages.

- The current example solely focuses on stacking an excessive number of models, but it would be beneficial to explore what seems to be a statistically sound approach that combines information from fewer models in a more frugal manner (less diversity, fewer models).

In conclusion, the proposed methodology is interesting and offers a unique perspective. However, the paper's details, in my humble opinion, do not effectively convey the core content of the approach to the reader. Additionally, the choice of application does not solidify the confidence in the method for the user. I recommend to revisit the structure of the paper to really expose the core contribution.

Reviewer #2: A novel stacking strategy will be presented in this paper. This new strategy results from reinterpreting the model selection process based on the generalization error. For the first time, it will be demonstrated that this problem can be translated into a well-studied financial problem: portfolio management and optimization. The subject addressed is interesting and within the scope of the PONE. After solving the following issues, the manuscript could be possible for publication.

- Authors should add quantity results to the abstract.

- The literature review is too old, new studies should be reviewed. You can cite to following studies:

Morshed-Bozorgdel, A., Kadkhodazadeh, M., Valikhan Anaraki, M., & Farzin, S. (2022). A novel framework based on the stacking ensemble machine learning (SEML) method: application in wind speed modeling. Atmosphere, 13(5), 758.

Shi, J., Li, C., & Yan, X. (2023). Artificial intelligence for load forecasting: A stacking learning approach based on ensemble diversity regularization. Energy, 262, 125295.

- The structure of the article is not interesting. An article should include the parts of the abstract, introduction, materials and methods, results and discussion, and conclusion. This study should be revised according to the mentioned structure.

- The statistical characteristics of case studies should be presented in one table.

- More figures such as the convergence curve of optimization algorithms should be added to the manuscript.

- Are parameters of the introduced algorithm selected by sensitivity analysis? Please add the results of the sensitivity analysis.

- Authors should add quantity results in the conclusion.

6. PLOS authors have the option to publish the peer review history of their article (what does this mean?). If published, this will include your full peer review and any attached files.

Reviewer #1: No

Reviewer #2: **Yes: **Dr Saeed Farzin

---

## [Author Response · Author response to Decision Letter 0]

1 Aug 2023

Please check the "Response to Reviewes of..." file.

---

## [Editor Report · Decision Letter 1]

4 Aug 2023

HPOSS: A hierarchical portfolio optimization stacking strategy to reduce the generalization error of ensembles of models

PONE-D-23-08481R1

Dear Dr. Ozelim,

We’re pleased to inform you that your manuscript has been judged scientifically suitable for publication and will be formally accepted for publication once it meets all outstanding technical requirements.

Kind regards,

Argimiro Arratia, Ph.D.

Academic Editor

PLOS ONE

Additional Editor Comments (optional):

The paper is now well structured and the goals are clear and well argued. As a final advice I strongly recommend to read through the paper carefully to correct any typo. For example, in a quick final read I found the followings (please correct for the final version)

line 559: the the

in references 32. STONE should be Stone

52. the title is all in cap letters, should capitalize only the first letter of each word
---

## [Editor Report · Acceptance letter]

21 Aug 2023

PONE-D-23-08481R1 

HPOSS: A hierarchical portfolio optimization stacking strategy to reduce the generalization error of ensembles of models 

Dear Dr. Ozelim:

I'm pleased to inform you that your manuscript has been deemed suitable for publication in PLOS ONE. Congratulations! Your manuscript is now with our production department. 

Kind regards, 

on behalf of

Dr. Argimiro Arratia 

Academic Editor

PLOS ONE